# Structural basis of human transcription–DNA repair coupling

Goran Kokic[1], Felix R. Wagner[1], Aleksandar Chernev[2], Henning Urlaub[2,3] & Patrick Cramer[1✉]

Transcription-coupled DNA repair removes bulky DNA lesions from the genome[1,2] and protects cells against ultraviolet (UV) irradiation[3]. Transcription-coupled DNA repair begins when RNA polymerase II (Pol II) stalls at a DNA lesion and recruits the Cockayne syndrome protein CSB, the E3 ubiquitin ligase, CRL4[CSA] and UV-stimulated scaffold protein A (UVSSA)[3]. Here we provide five high-resolution structures of Pol II transcription complexes containing human transcription-coupled DNA repair factors and the elongation factors PAF1 complex (PAF) and SPT6. Together with biochemical and published[3,4] data, the structures provide a model for transcription–repair coupling. Stalling of Pol II at a DNA lesion triggers replacement of the elongation factor DSIF by CSB, which binds to PAF and moves upstream DNA to SPT6. The resulting elongation complex, EC[TCR], uses the CSA-stimulated translocase activity of CSB to pull on upstream DNA and push Pol II forward. If the lesion cannot be bypassed, CRL4[CSA] spans over the Pol II clamp and ubiquitylates the RPB1 residue K1268, enabling recruitment of TFIIH to UVSSA and DNA repair. Conformational changes in CRL4[CSA] lead to ubiquitylation of CSB and to release of transcription-coupled DNA repair factors before transcription may continue over repaired DNA.

Eukaryotic cells use transcription-coupled DNA repair (TCR) to eliminate bulky DNA lesions, such as UV light-induced pyrimidine dimers[1,2]. When Pol II encounters a bulky lesion in the template strand, transcription elongation stalls and this triggers TCR[5]. TCR requires the Cockayne syndrome proteins, CSB and CSA, and UVSSA[3,4]. CSB, CSA and UVSSA bind to arrested Pol II in vivo[4,6,7] and recruit the DNA nucleotide excision repair machinery[3]. CSB binds to the Pol II elongation complex[8–10] and to CSA[11,12], which in turn binds to UVSSA[4]. CSA also associates with DNA damage-binding protein 1 (DDB1), cullin-4A (CUL4A) and RING box protein 1 (RBX1) to form CRL4[CSA], an E3 ubiquitin ligase[13,14]. CRL4[CSA] may be responsible for ubiquitylation of damage-stalled Pol II at RPB1 residue K1268, which is required for TCR and transcription restart[4,7,15,16]. Following ubiquitylation of Pol II, UVSSA recruits TFIIH and other factors that are required for nucleotide excision repair[4,7,17].

As a first step towards understanding the molecular mechanism of TCR, the structural basis for Pol II stalling at a pyrimidine dimer lesion was previously reported[18]. Another study has localized the yeast counterpart of CSB (Rad26) on the Pol II elongation complex and suggested that its translocase activity pulls on upstream DNA to push Pol II onto the lesion[19]. However, yeast lacks counterparts of CSA and UVSSA, prompting us to study the human TCR mechanism. This is feasible based on the structure of the Pol II elongation complex EC*, which contains the elongation factors DSIF[20], PAF, RTF1 and SPT6 (ref. [21]). Using cryo-electron microscopy (cryo-EM), we resolved five different Pol II elongation complex structures containing CSB, the CSA–DDB1 complex or the complete CRL4[CSA], and UVSSA. Together with biochemical probing, our results provide the structural basis for coupling transcription to DNA repair in human cells.

## Structure of Pol II–TCR complex

We prepared recombinant human CSB, the CSA–DDB1 complex and UVSSA and tested the effect of these factors on Pol II transcription over an arrest sequence (Extended Data Fig. 1a). CSB facilitated Pol II passage over the arrest sequence, as previously described[8,19]. UVSSA also facilitated Pol II passage to some extent, whereas CSA–DDB1 did not. When all three factors were present, a more than additive effect of stimulation by CSB and UVSSA was observed, indicating that TCR factors cooperatively stimulate Pol II elongation. This effect was largely due to stimulation of the ATPase activity of CSB by CSA (Extended Data Fig. 1b, c).

Consistent with these results, our recombinant TCR factors bound a Pol II elongation complex containing a large DNA bubble[19] (Extended Data Fig. 1e). We solved the structure of the resulting Pol II–CSB–CSA–DDB1–UVSSA elongation complex (referred to here as the Pol II–TCR complex) by cryo-EM at an overall resolution of 2.8 Å (structure 1; Fig. 1a, Extended Data Figs. 3, 10). UVSSA was more flexible, but helical density features and crosslinking mass spectrometry data enabled unambiguous placement of a homology model for the N-terminal Vsp27–Hrs–STAM (VHS) domain[4,22,23] (Extended Data Figs. 2a, 10i, j; Methods). This resulted in a nearly complete atomic model of the Pol II–TCR complex with very good stereochemistry (Extended Data Table 1).

The structure of the TCR complex shows that CSB contacts upstream DNA, UVSSA lies near downstream DNA, and CSA forms a bridge between them (Fig. 1, Supplementary Video 1). The binding of CSB alters the trajectory of upstream DNA by approximately 50°, essentially as observed for Rad26 (ref. [19]) (Fig. 1a). CSB contacts the

[1]Department of Molecular Biology, Max Planck Institute for Biophysical Chemistry, Göttingen, Germany. [2]Max Planck Institute for Biophysical Chemistry, Bioanalytical Mass Spectrometry, Göttingen, Germany. [3]University Medical Center Göttingen, Institute of Clinical Chemistry, Bioanalytics Group, Göttingen, Germany. ✉e-mail: patrick.cramer@mpibpc.mpg.de

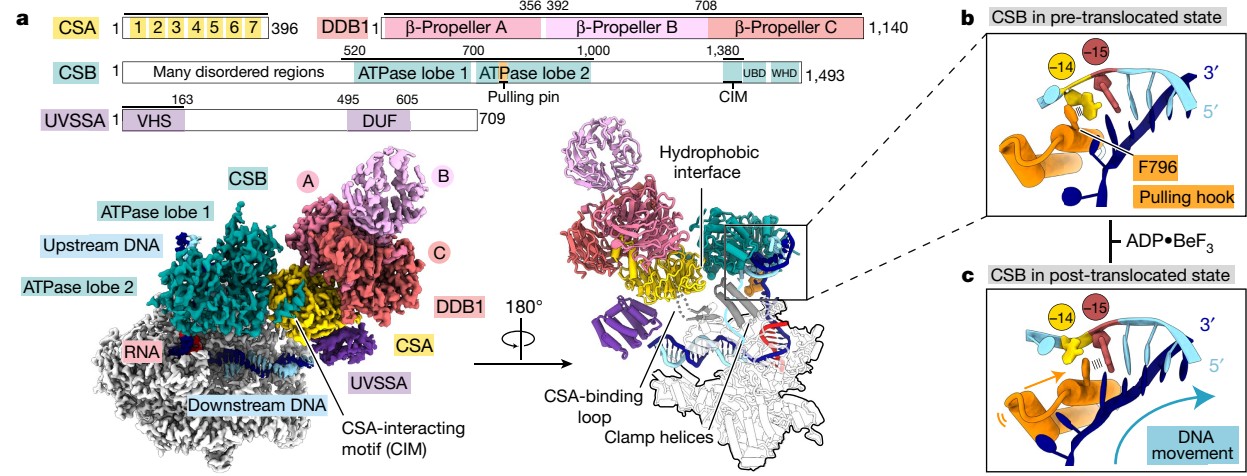

**Fig. 1 | Structure of the Pol II–TCR complex. a**, Cryo-EM density (left) and ribbon model (right) of the Pol II–CSB–CSA–DDB1–UVSSA complex. The scheme depicts the domain composition and colour code for proteins. The solid black lines mark residues included in the model.

UBD, ubiquitin-binding domain; WHD, winged-helix domain; DUF, domain of unknown function. **b**, Zoom-in on the upstream DNA fork bound to the pulling hook of CSB in the pre-translocated state. **c**, Same as in **b** but in the post-translocated state.

Pol II clamp and protrusion that form opposite sides of the active centre cleft (Fig. 1a). CSA and UVSSA do not bind to Pol II, consistent with their recruitment by CSB in vivo[4]. CSB binds to the CSA β-propeller domain with its ATPase lobe 1 and with its helical CSA-interacting motif (Fig. 1a, Extended Data Fig. 2d), which is essential for CSA binding and TCR in vivo[4]. The binding of the CSA-interacting motif to CSA might restrict the mobile ATPase lobe 2 of CSB to an active conformation, explaining the stimulation of CSB ATPase activity by CSA (Extended Data Fig. 6e). UVSSA contacts the opposite face of the CSA β-propeller via its VHS domain (Fig. 1a, Extended Data Fig. 10i, j), which explains why recruitment of UVSSA to stalled Pol II depends on CSA in vivo[4,23,24].

The structure rationalizes known mutations associated with Cockayne syndrome[25] (Extended Data Fig. 4). Many mutations in CSB and CSA cluster in the CSB–CSA interface, including three CSB mutations (R670W, W686C and S687L) that lead to severe type II Cockayne syndrome. Other mutations in CSB are found in the ATP-binding site and in the interface with upstream DNA and probably impair CSB function. By contrast, the CSA mutation W361C, which causes UV-sensitive syndrome[25], maps to the CSA–UVSSA interface and is predicted to impair recruitment of UVSSA. This supports the view that recruitment of UVSSA is critical for transcription–repair coupling, whereas loss of CSB and CSA might additionally impair transcription elongation or processing of stalled Pol II, perhaps explaining the more-severe clinical manifestations of Cockayne syndrome than of UV-sensitive syndrome[3,25].

## CSB translocase and elongation stimulation

Our structure suggests how the translocase activity of CSB pushes Pol II forward. The two ATPase lobes hold upstream DNA, and a helix–loop–helix element ('pulling hook') protrudes from lobe 2 and inserts into the upstream fork of the DNA bubble (Fig. 1b, Extended Data Fig. 6a, c, d). The pulling hook contains a conserved phenylalanine residue (F796) that stacks with its aromatic side chain against the first base of the non-template strand at position –14 (Fig 1b, Extended Data Fig. 6a). Substitution of F796 by alanine impairs CSB activity, showing that the pulling hook is required for CSB function (Extended Data Fig. 1f, g). Rad26 contains an element corresponding to the pulling hook[19] (Extended Data Fig. 6d).

To better understand the translocase mechanism of CSB, we also solved the Pol II–TCR structure in the presence of ADP•BeF₃ at 2.7 Å

resolution (structure 2; Extended Data Figs. 5, 10). The CSB translocase adopts the pre-translocated and post-translocated states in structures 1 and 2, respectively, elucidating the mechanism of the translocase (Fig. 1c, Extended Data Fig. 6c, Supplementary Video 2). Upon ATP binding, the ATPase lobe 2 of CSB closes and pulls on the template strand of the upstream DNA, whereas the pulling hook pulls on the non-template strand in the same direction. As lobe 1 of CSB is anchored to the Pol II clamp, pulling on the upstream DNA template will push Pol II forward.

## Switching from elongation to TCR

Comparison of the Pol II–TCR structure with the structure of the Pol II EC* (ref. [21]) reveals a clash between CSB and DSIF, which comprises the subunits SPT4 and SPT5 (ref. [20]). We therefore tested whether the binding of CSB to Pol II displaces DSIF. We labelled DNA, CSB and DSIF with different fluorescent dyes and monitored the composition of Pol II complexes by electrophoretic mobility shift assay (Methods). The addition of increasing amounts of CSB indeed displaced DSIF from Pol II (Fig. 2a). To investigate whether CSB could replace DSIF during transcription, we conducted RNA elongation assays (Fig. 2b). Whereas the Pol II–DSIF complex could not transcribe over an arrest sequence, the addition of CSB stimulated the passage of Pol II, indicating that CSB replaced DSIF on transcribing Pol II. Competition between DSIF and CSB for Pol II binding explains the observation that SPT5 can repress TCR[26]. In summary, these results indicate that the switch from active Pol II elongation to TCR involves replacement of DSIF by CSB on the surface of Pol II.

To further investigate the switch from elongation to TCR, we extended our Pol II–TCR complex preparation by adding SPT6, PAF and RTF1, and analysed the resulting complex by cryo-EM (Extended Data Fig. 7). The overall resolution of the structure was 2.9 Å and it revealed all factors except RTF1, which probably dissociated after loss of DSIF (structure 3; Fig. 2c, Extended Data Fig. 8a). This 22-subunit Pol II–CSB–CSA–DDB1–UVSSA–SPT6–PAF complex represents an alternative elongation complex that we call EC^TCR.

The conversion of EC* to EC^TCR involves three structural changes (Supplementary Video 3). First, TCR factors replace DSIF and displace RTF1, which interacts with DSIF[21,27]. Second, upstream DNA moves and contacts the helix–hairpin–helix domain of SPT6 (Fig. 2c). Helix–hairpin–helix domains are found in several DNA-binding

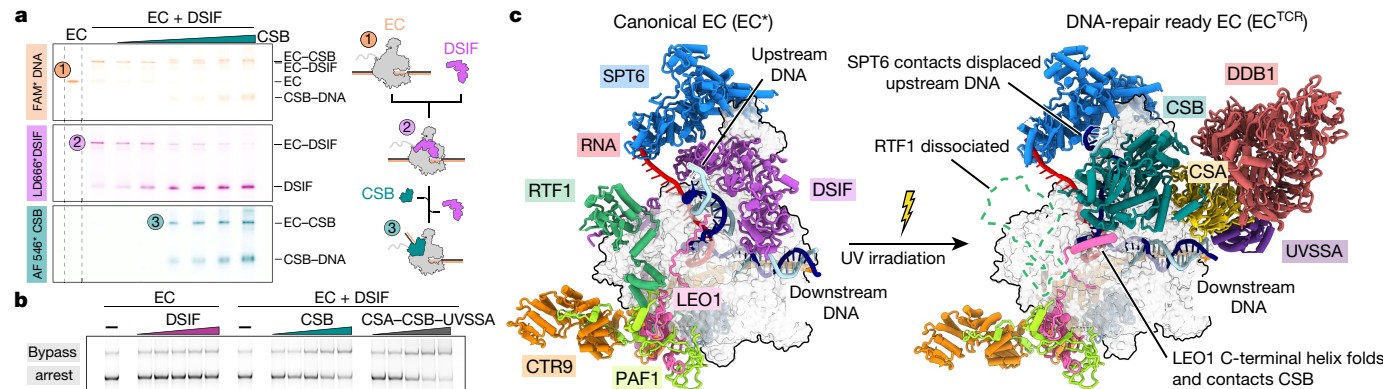

**Fig. 2 | Formation and structure of EC^TCR. a**, Electrophoretic mobility shift assay monitors replacement of DSIF by CSB on the Pol II elongation complex. The gel was scanned in three different channels to reveal the elongation complex (via DNA), DSIF and CSB through different fluorescent labels. The experiment was repeated three times. For gel source data, see Supplementary Fig. 1. **b**, CSB can replace DSIF during elongation and stimulate Pol II progression over an arrest site. The experiment was repeated three times. For gel source data, see Supplementary Fig. 1. **c**, Differences between EC* (left; PDB code: 6TED[21]) and the new EC^TCR (right).

proteins[28] and yeast Spt6 binds to DNA[29]. Modelling shows that extension of upstream DNA leads to a clash with parts of SPT6, and thus changes in the position of DNA or SPT6 are required to accommodate longer DNA (Extended Data Fig 8b). Third, the C-terminal linker of the PAF subunit, LEO1, that contacts upstream DNA in EC* (ref. [30]) moves by up to approximately 30 Å and forms a helix that binds to lobe 2 of the CSB ATPase (Fig. 2c). This contact accounts for the known PAF–CSB interaction that is induced by UV light in vivo[31,32] and for our observation that PAF stimulates CSB ATPase activity (Extended Data Fig. 1d, h).

## Ubiquitylation by CRL4^CSA

The cellular response to UV irradiation not only involves recruitment of TCR factors but also ubiquitylation events[7,12,16,32]. Ubiquitylation of the largest Pol II subunit, RPB1, on K1268 regulates transcription shutdown and recovery[7,16]. In addition, the E3 ligase CRL4^CSA polyubiquitylates CSB, leading to degradation of CSB[12]. To investigate these events in vitro, we performed ubiquitylation assays with the complete Pol II–TCR complex containing CRL4^CSA (Fig. 3a). We observed ubiquitylation of CSA and CUL4A and polyubiquitylation of CSB, as previously described[14]. We also detected ubiquitylation of UVSSA and

identified 11 ubiquitylation sites on RPB1, including residue K1268 as the highest-scoring site (Fig. 3b). Ubiquitylation of Pol II was dependent on CSB and occurred in the absence of UVSSA (Extended Data Fig 9a), as shown in vivo[33]. These results indicate that CRL4^CSA is the E3 ligase that ubiquitylates K1268.

We then solved the structure of the complete Pol II–TCR complex including CRL4^CSA at an overall resolution of 3.0 Å (Fig. 3c, Extended Data Figs. 9, 10). Focused classification revealed two distinct states that differed in the conformation of CRL4^CSA (Extended Data Figs. 9, 10). In the first state (structure 4), UVSSA positions the C-terminal domain of CUL4A such that RBX1 faces a loop in the RPB1 jaw domain that contains the ubiquitylated residue K1268. RBX1 binds to an E2 enzyme–ubiquitin complex[34], which we modelled onto our structure (Fig. 3c). In this model, the activated C terminus of ubiquitin is positioned at the K1268-containing loop, which explains how CRL4^CSA directs site-specific Pol II ubiquitylation.

In the second state of the complete Pol II–TCR complex structure (structure 5), CUL4A and RBX1 have moved over a large distance to reach the C-terminal region of CSB (Fig. 3c). This CSB region is targeted by ubiquitylation[35] and is essential for TCR[36]. Conversion of the first state to the second state of the complete EC^TCR complex requires extensive rearrangements within CRL4^CSA, including an

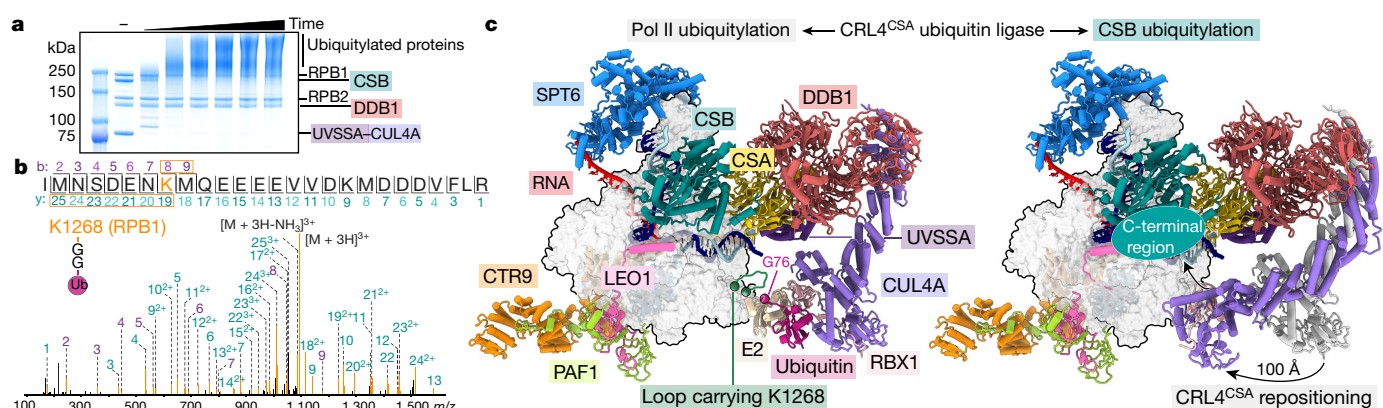

**Fig. 3 | Complete Pol II–TCR complex and ubiquitylation by CRL4^CSA. a**, In vitro ubiquitylation of the complete Pol II–TCR complex by CRL4^CSA. The experiment was repeated two times. For gel source data, see Supplementary Fig. 1. **b**, Tandem mass spectrometry fragment spectrum of the RPB1 peptide 1261-IMNSDENK(Gly-Gly)MQEEEE VVDKMDDDVFLR-1286. **c**, Structure of the EC^TCR containing CRL4^CSA. Two conformations related to targeting of the Pol II jaw domain (left) or CSB (right) for ubiquitylation are shown. RBX1 and the E2 enzyme–donor ubiquitin complex were modelled as described in the Methods.

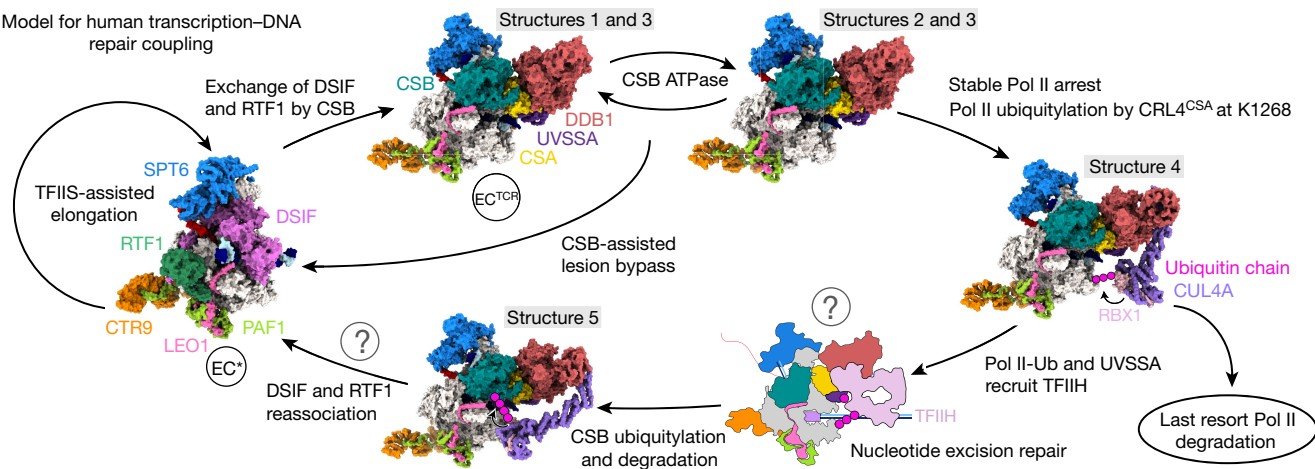

**Fig. 4 | Model for human transcription–DNA repair coupling.** The cycle starts from EC* (left) and involves several intermediate states of EC^TCR that we structurally define here. The question marks denote uncertain parts of the model, such as exposure of the DNA lesion and resumption of transcription following DNA repair.

approximately 70° rotation of the DDB1 β-propeller B and a roughly 100 Å displacement and approximately 20° rotation of CUL4A (Supplementary Video 4). Modelling an E2 enzyme–ubiquitin complex onto structure 5 reveals minor clashes with Pol II, showing that some adjustments are required for E2 binding. The conversion to the second state may occur when nucleotide excision repair factors are recruited to downstream DNA. Thus, stable conversion of the complete EC^TCR complex to the state observed in structure 5 is predicted to enable ubiquitylation of CSB and to complete transcription–repair coupling.

## TCR model

Our work converges with published data[3] on a molecular model that explains how the complete TCR complex, consisting of CSB, CRL4^CSA and UVSSA, mechanistically couples transcription to DNA repair (Fig. 4). When Pol II encounters an obstacle that cannot be overcome with the elongation factor TFIIS, Pol II stalls and the TCR complex binds. This requires displacement of DSIF and converts EC* to EC^TCR, which then uses the ATPase activity of CSB to push Pol II forward. If the obstacle can be bypassed, Pol II resumes elongation and EC* is re-established. If the obstacle cannot be overcome, CRL4^CSA ubiquitylates Pol II at K1268, leading to recruitment of TFIIH near UVSSA[4] and downstream DNA. TFIIH may then use ATPase activity[17] to push Pol II backwards[37] and enable DNA repair. Finally, rearrangement of CRL4^CSA leads to polyubiquitylation of CSB and degradation by the proteasome, which releases TCR factors that are all anchored via CSB. Release of TCR factors liberates the Pol II sites for DSIF and RTF1, and EC* is re-established. Pol II may resume transcription after DNA repair. Interconversion between EC* and EC^TCR may facilitate both bypass of small DNA lesions and repair of large lesions. Alternatively, Pol II becomes persistently stalled and is degraded[38].

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

# Methods

No statistical methods were used to predetermine sample size. The experiments were not randomized, and the investigators were not blinded to allocation during experiments and outcome assessment.

## Cloning and protein expression

Vectors encoding full-length human CSA, DDB1, CSB, UVSSA, CUL4A and mouse RBX1 were obtained from Harvard Medical School PlasmID Repository. Genes were amplified by PCR and cloned into respective vectors by ligation-independent cloning[39]. CSA and RBX1 were cloned into the 438A vector (addgene no. 55218), and CSB, DDB1, UVSSA and CUL4A were cloned into the 438B vector (addgene no. 55219), resulting in no tag or 6×His tag, respectively. CSA and DDB1, and CSA, DDB1, CUL4A and RBX1 were combined into single vectors by ligation-independent cloning[39]. The CSB ATPase-deficient mutant (CSB K538R)[40] and the pulling hook mutant (CSB F796A) were produced by around-the-horn-mutagenesis and expressed and purified as their wild-type counterparts. For fluorescent labelling of CSB, the gene was cloned into the 438-SNAP-V1 vector (addgene no. 55222), which resulted in a SNAPf tag at the N terminus. For fluorescent labelling of DSIF, a ybbR tag[41] preceded by a GGGG linker was introduced to the C terminus of Spt4 by around-the-horn mutagenesis.

Proteins were expressed in insect cells. Sf9 (ThermoFisher), Sf21 (Expression Systems) and Hi5 (Expression Systems) cell lines were not tested for mycoplasma contamination and were not authenticated in-house. Preparation of bacmids and baculoviruses has previously been described in detail[42]. In brief, 600 ml of Hi5 cells grown in ESF-921 medium were infected with V1 virus and grown for 2–3 days. Cells were collected by centrifugation (30 min, 4 °C, 500$g$) and resuspended in lysis buffer (400 mM NaCl, 20 mM Tris-HCl pH 7.9, 10% glycerol (v/v), 1 mM DTT, 30 mM imidazole pH 8.0, 0.284 μg ml$^{-1}$ leupeptin, 1.37 μg ml$^{-1}$ pepstatin A, 0.17 mg ml$^{-1}$ PMSF and 0.33 mg ml$^{-1}$ benzamidine). Cell suspension was frozen in liquid nitrogen and stored at −80 °C until protein purification.

## Protein purification

Pol II was purified from the pig thymus as previously described[30,43]. Human transcription elongation factors (DSIF, PAF, SPT6, RTF1 and P-TEFb) were prepared as previously described[20,21,30]. All protein purification steps were performed at 4 °C, unless stated otherwise. The purity of protein preparations was monitored by SDS–PAGE using NuPAGE 4–12% Bis-Tris protein gels (Invitrogen), followed by Coomassie staining. Initial purification steps were the same for all TCR proteins. Cells were thawed in a water bath at 30 °C. Cells were opened by sonication and the lysate was clarified by centrifugation and ultracentrifugation. Clarified lysate was further filtrated through 0.8-μm syringe filters and applied onto a HisTrap HP 5-ml column (GE Healthcare) equilibrated in lysis buffer. The column was washed with 5 CV of lysis buffer, 20 CV of high-salt buffer (800 mM NaCl, 20 mM Tris-HCl pH 7.9, 10% glycerol (v/v), 1 mM DTT, 30 mM imidazole pH 8.0, 0.284 μg ml$^{-1}$ leupeptin, 1.37 μg ml$^{-1}$ pepstatin A, 0.17 mg ml$^{-1}$ PMSF and 0.33 mg ml$^{-1}$ benzamidine) and again 5 CV lysis buffer. Proteins were eluted with a 0–80% gradient of elution buffer (400 mM NaCl, 20 mM Tris-HCl pH 7.9, 10% glycerol (v/v), 1 mM DTT, 500 mM imidazole pH 8.0, 0.284 μg ml$^{-1}$ leupeptin, 1.37 μg ml$^{-1}$ pepstatin A, 0.17 mg ml$^{-1}$ PMSF and 0.33 mg ml$^{-1}$ benzamidine). In the case of the CSA–DDB1 complex, an additional step was introduced at this point to separate the CSA–DDB1 complex from excess of DDB1. After a high-salt wash, the column was washed with a low-salt buffer (150 mM NaCl, 20 mM Tris-HCl pH 7.9, 10% glycerol (v/v), 1 mM DTT, 30 mM imidazole pH 8.0, 0.284 μg ml$^{-1}$ leupeptin, 1.37 μg ml$^{-1}$ pepstatin A, 0.17 mg ml$^{-1}$ PMSF and 0.33 mg ml$^{-1}$ benzamidine) and protein was eluted directly onto a 5-ml HiTrapQ HP column (GE Healthcare) in a low-salt elution buffer (150 mM NaCl, 20 mM Tris-HCl pH 7.9, 10% glycerol (v/v), 1 mM DTT, 500 mM imidazole pH 8.0, 0.284 μg ml$^{-1}$

leupeptin, 1.37 μg ml$^{-1}$ pepstatin A, 0.17 mg ml$^{-1}$ PMSF and 0.33 mg ml$^{-1}$ benzamidine). The HiTrapQ column was washed with 5 CV of low-salt buffer and proteins were eluted with a 0–100% of monoQ elution buffer (1 M NaCl, 20 mM Tris-HCl pH 7.9, 10% glycerol (v/v), 1 mM DTT, 500 mM imidazole pH 8.0, 0.284  μg ml$^{-1}$ leupeptin, 1.37 μg ml$^{-1}$ pepstatin A, 0.17 mg ml$^{-1}$ PMSF and 0.33 mg ml$^{-1}$ benzamidine).

For all TCR proteins, appropriate protein fractions were pulled, mixed with 2 mg of TEV protease and dialysed overnight against the dialysis buffer (400 mM NaCl, 20 mM Tris-HCl pH 7.9, 10% glycerol (v/v) and 1 mM DTT). After, dialysis protein solution was passed through a 5-ml HisTrap column equilibrated in dialysis buffer. Flow-through containing the protein was collected, concentrated and loaded onto Superdex 200 10/300 increase column (GE Healthcare) equilibrated in storage buffer (400 mM NaCl, 20 mM NaOH:HEPES pH 7.5, 10% glycerol (v/v) and 1 mM DTT). Peak fractions were pulled, concentrated, flash frozen and stored at −80 °C.

CSB containing N-terminal SNAPf and TwinStrepII tag was purified as follows. The clarified lysate was incubated with 1 ml of Strep-TactinXT 4Flow high-capacity resin (IBA) pre-equilibrated in lysis buffer and washed extensively with lysis buffer. Protein was eluted with BXT buffer (IBA), concentrated and loaded onto Superdex 200 10/300 increase column (GE Healthcare) equilibrated in storage buffer (400 mM NaCl, 20 mM NaOH:HEPES pH 7.5, 10% glycerol (v/v) and 1 mM DTT). Peak fractions were pulled, concentrated, flash frozen and stored at −80 °C.

## RNA extension assays

DNA and RNA oligonucleotides were ordered from Integrated DNA Technologies. Sequences[19] used in the assay are: CTA CAT ACA CCA CAC ACC ACA CCG AGA AAA AAA AAT TAC CCC TTC ACC CTC ACT GCC CCA CAT TCT AAC CAC ACA TCA CTT ACC TGG ATA CAC CCT TAC TCC TCT CGA TAC CTC ACC ACC TTA CCT ACC ACC CAC (template strand); GTG GGT GGT AGG TAA GGT GGT GAG GTA TCG AGA GGA GTA AGG GTG TAT CCA GGT AAG TGA TGT GTG GTT AGA ATG TGG GGC AGT GAG GGT GAA GGG GTA ATT TTT TTT TCT CGG TGT GGT GTG TGG TGT ATG TAG (non-template strand); and /5Cy5/rUrUrA rUrUrArU rUrUrUrU rArUrU rCrUrU rArUrC rGrA rGrArG rGrA (RNA). Template-strand DNA and RNA were mixed in equimolar ratio and annealed in water by heating to solution to 95 °C followed by slow cooling (1 °C per min) to 4 °C. Pol II was mixed with DNA–RNA scaffold in equimolar ratio and incubated at 30 °C for 10 min. Next, 1.5 M excess of non-template DNA was added and the solution was incubated for 10 min more at 30 °C. A typical RNA extension reaction contained Pol II (200 nM) in the final buffer containing 100 mM NaCl, 20 mM HEPES pH 7.5, 5% (v/v) glycerol, 5 mM MgCl$_2$ and 1 mM DTT. When proteins were titrated, the highest protein concentration was 2 μM (in case of a protein mixture, concentration of each factor was 2 μM), followed by a half-log dilution series. In the case of the DSIF–CSB competition assay, Pol II was pre-incubated with 1.5× excess of DSIF before addition of TCR factors. Reactions were pre-incubated at 37 °C for 5 min and started with the addition of NTPs (0.5 mM GTP, UTP and CTP, 1 mM ATP and 0.5 mM dATP). Reactions were quenched with 2× quenching buffer (7 M urea in TBE buffer, 20 mM EDTA and 10 μg ml$^{-1}$ proteinase K (Thermo Scientific)). Proteins were digested for 30 min at 37 °C. RNA products were separated on a sequencing gel and visualized with a Typhoon FLA 9500 (GE Healthcare Life Sciences). Gel quantification was performed with ImageJ software and data were plotted with Prism 9 software.

## Three-colour electrophoretic mobility shift assay

DNA and RNA oligonucleotides were ordered from Integrated DNA Technologies. Sequences[19] used in the assay are: /56-FAM/CGC TCT GCT CCT TCT CCC ATC CTC TCG ATG GCT ATG AGA TCA ACT AG (template strand); CTA GTT GAT CTC ATA GCC ATC GAG AGG ATG GGA GAA GGA GCA GAG CG (non-template strand); and rArCrA rUrCrA rUrArA rCrArU rUrUrG rArArC rArArG rArArU rArUrA rUrUrU rArCrA rArArA rUrCrG

rArGrA rGrGrA (RNA). For this assay, CSB and DSIF were fluorescently labelled. SNAPf–CSB (50 μM) was incubated with 10× molar excess of SNAP-Surface 546 substrate (New England BioLabs) overnight at 4 °C in CSB storage buffer. Labelled CSB was purified from the excess dye by Superdex 200 10/300 increase column (GE Healthcare) equilibrated in storage buffer (400 mM NaCl, 20 mM NaOH:HEPES pH 7.5, 10% glycerol (v/v) and 1 mM DTT). Labelling efficiency was around 100%. DSIF subunit SPT4 contained a ybbR tag on the C terminus and the protein was labelled by using Sfp phosphopantetheinyl transferase, as previously described in detail[41]. Substrate for the labelling reaction was LD666-CoA (Lumidyne Technologies) and the labelling efficiency was around 85%. The Pol II elongation complex was assembled by incubating Pol II with 1.3× excess of template strand:RNA for 10 min at 30 °C, followed by the addition of 1.5× excess of non-template strand and further incubation for 10 min at 30 °C. Next, the Pol II elongation complex was supplemented with 1.2× excess of DSIF and incubated for 10 min at 30 °C. Finally, CSB was titrated in the reaction and the reaction was further incubated for 10 min at 30 °C. Final reaction contained Pol II (100 nM), DSIF (120 nM) and CSB (400 nM, 200 nM, 150 nM, 100 nM, 50 nM and 25 nM) in final buffer containing 100 mM NaCl, 20 mM HEPES pH 7.5, 10% glycerol, 2 mM $MgCl_2$ and 1 mM DTT. Reactions were loaded on a NativePAGE 3–12% Bis-Tris gels (Thermo Scientific) and ran at 150 V for 1.5 h. The gels were scanned in Typhoon FLA 9500 (GE Healthcare Life Sciences) in three different channels for the visualization of template-strand DNA, CSB and DSIF.

## Analytical size-exclusion chromatography

Analytical size-exclusion chromatography was used to monitor association of TCR factors with Pol II (Extended Data Fig. 1e) and to monitor RTF1 association with EC* and EC$^{TCR}$ (Extended Data Fig. 8a). In the case of TCR factors, the proteins were mixed in equimolar ratios in the final size-exclusion buffer (100 mM NaCl, 20 mM HEPES 7.5, 5% glycerol, 1 mM $MgCl_2$ and 1 mM DTT) and ran over a Superose 6 Increase 3.2/300 column. The Pol II elongation complex was formed as for structure 1. In the case of RTF1 binding, all factors were added to the pre-formed Pol II elongation complex in 1.5× excess in the final size-exclusion buffer and incubated for 1 h at 30 °C in the presence of 1 mM ATP and P-TEFb. The complexes were injected onto a Superose 6 Increase 3.2/300 column and the fractions were analysed by SDS–PAGE. The template strand and RNA used for the EC* and EC$^{TCR}$ formation were the same (template strand: CGC TCT GCT CCT TCT CCC ATC CTC TCG ATG GCT ATG AGA TCA ACT AG; RNA: rArCrA rUrCrA rUrArA rCrArU rUrUrG rArArC rArArG rArArU rArUrA rUrArU rArCrA rArArA rUrCrG rArGrA rGrGrA) but differed in the non-template strand, which was fully complementary to the template strand in the case of EC* (non-template strand: CTA GTT GAT CTC ATA GCC ATC GAG AGG ATG GGA GAA GGA GCA GAG CG) or formed a large bubble with the template strand in the case of EC$^{TCR}$ (non-template strand: CTA GTT GAT CTC ATA TTT CAT TCC TAC TCA GGA GAA GGA GCA GAG CG).

## In vitro ubiquitylation assay

Ubiquitin, UBE1 and UbcH5b were purchased from Boston Biochem. The Pol II elongation complex was formed as for structural analysis of structure 1. The ubiquitylation reaction contained Pol II ECs (0.8 μM), CSB (0.8 μM), UVSSA (0.8 μM), CSA–DDB1–CUL4A–RBX1 (0.8 μM), UBE1 (150 nM), UbcH5b (0.5 μM) and ubiquitin (300 μM) in 100 mM NaCl, 50 mM Tris pH 7.9, 10 mM $MgCl_2$, 0.2 mM $CaCl_2$, 5% glycerol and 1 mM DTT. Reactions were started by the addition of ATP (3 mM) and stopped with EDTA (15 mM). Proteins were separated on NuPAGE 4–12% Bis-Tris protein gels (Invitrogen) and stained with Coomassie. In the case of the ubiquitylation assay in the absence of CSB or UVSSA, the assay was performed as described above, but with lower concentrations of Pol II, CRL4$^{CSA}$ and CSB or UVSSA (0.4 μM). The proteins were separated on 3–8% Tris-acetate gel (Invitrogen) and transferred onto a PVDF membrane with a Trans-Blot Turbo Transfer System (Bio-Rad)

for immunoblotting. The membrane was blocked with 5% (w/v) milk powder in PBS containing 0.1% Tween-20 (PBST) for 1 h at room temperature. The membrane was then incubated with F-12 anti-RPB1 antibody (1:100 dilution; Santa Cruz Biotechnology) in PBST supplemented with 2.5% (w/v) milk powder. After washing the membrane with PBST, the membrane was incubated with an anti-mouse HRP conjugate (1:3,000 dilution; ab5870, Abcam) in PBST supplemented with 1% (w/v) milk powder for 1 h at room temperature. The membrane was developed with SuperSignal West Pico Chemiluminescent Substrate (Thermo Fisher) and scanned with a ChemoCam Advanced Fluorescence imaging system (Intas Science Imaging).

## ATPase assay

The enzyme-coupled ATPase assay uses two separate fast enzymatic reactions to couple ATP regeneration to NADH oxidation. The typical reaction contained 100 nM protein in buffer containing 50 mM potassium acetate, 20 mM KOH-HEPES pH 7, 5 mM magnesium acetate, 5% glycerol (v/v), 0.2 mg ml$^{-1}$ BSA, 3 mM phosphoenolpyruvate (PEP), 0.3 mM NADH and excess pyruvate kinase and lactate dehydrogenase enzyme mix (Sigma). The reaction mixture was incubated for 10 min at 30 °C and the reaction was started by addition of ATP (1.5 mM final). The rate of ATP hydrolysis was monitored by measuring a decrease in the absorption at 340 nm using the Infinite M1000Pro reader (Tecan). Resulting curves were fit to a linear model using GraphPad Prism version 9.

## Crosslinking mass spectrometry

The Pol II elongation complex was formed as described in the RNA extension assay. DNA and RNA sequences used for elongation complex formation are the following[19]: CGC TCT GCT CCT TCT CCC ATC CTC TCG ATG GCT ATG AGA TCA ACT AG (template strand); CTA GTT GAT CTC ATA TTT CAT TCC TAC TCA GGA GAA GGA GCA GAG CG (non-template strand); and rArUrC rGrAr GrArG rGrA (RNA). Equimolar amounts of elongation complex, CSB, CSA–DDB1 and UVSSA were mixed in the final complex formation buffer of 100 mM NaCl, 20 mM HEPES pH 7.5, 1 mM DTT, 1 mM $MgCl_2$ and 5% glycerol. The complex was incubated at 30 °C for 10 min and subsequently purified over a Superose 6 Increase 3.2/300 column equilibrated in complex formation buffer. For BS3 crosslinking, the protein solution was supplemented with 1 mM BS3 and incubated at 30 °C for 30 min. The crosslinking was quenched with 50 mM ammonium bicarbonate. For EDC crosslinking, the complex formation buffer contained HEPES pH 6.7 instead of pH 7.5. The protein solution was supplemented with 2 mM EDC and 5 mM sulfo-NHS and incubated at 30 °C for 30 min. The crosslinking reaction was quenched with 50 mM 2-mercaptoethanol and 20 mM Tris pH 7.9.

Analysis of crosslinked peptides was performed as previously described[17]. The crosslinked proteins were reduced with 10 mM DTT for 30 min at 37 °C and alkylated with 40 mM iodoacetamide for 30 min at 25 °C. Protein digestion was performed overnight in denaturing conditions (1 M urea) with 5 μg trypsin (Promega) at 37 °C. Formic acid (FA) and acetonitrile (ACN) were added to the digested samples to 0.1% (v/v) and 5% (v/v) final concentrations. Samples were purified with Sep-Pak C18 1cc 50 mg sorbent cartridge (Waters) by washing away salts and contaminants with 5% (v/v) ACN, 0.1% (v/v) FA and eluting bound peptides with 80% (v/v) ACN and 0.1% (v/v) FA. The extracted peptides were dried under vacuum and resuspended in 30 μl 30% (v/v) ACN and 0.1% (v/v) trifluoroacetic acid (TFA). Size separation of peptides was performed with a Superdex Peptide PC3.2/30 column (GE Healthcare) at flow rate of 50 μl min$^{-1}$ 30% (v/v) ACN and 0.1% (v/v) TFA. Fractions (100 μl) corresponding to elution volume 1.1–2 ml were collected, dried under vacuum and resuspended in 20 μl 2% (v/v) ACN and 0.05% (v/v) TFA.

Mass spectrometry analysis was performed on the Q Exactive HF-X Mass Spectrometer (Thermo Fisher Scientific) coupled with the Dionex UltiMate 3000 UHPLC system (Thermo Fisher Scientific). Online chromatographical separation was achieved with an in-house packed C18 column

(ReproSil-Pur 120 C18-AQ, 1.9-µm pore size, 75-µm inner diameter and 30 cm in length; Dr. Maisch). Samples were analysed as three 5-µl injections, separated on a 75-min gradient: flow rate of 300 nl min$^{-1}$; mobile phase A was 0.1% (v/v) FA; mobile phase B was 80% (v/v) ACN and 0.08% (v/v) ACN. The gradient was formed with an increase from 8%/12%/18% mobile phase B to 38%/42%/48% (depending on the fraction). MS1 acquisition was achieved with the following settings: resolution of 120,000; mass range of 380–1,580 $m/z$; injection time of 50 ms; and automatic gain control target of $1 \times 10^6$. MS2 fragment spectra were collected with dynamic exclusion of 10 s and varying normalized collision energy for the different injection replicates (28%/30%/28–32%) and the following settings: isolation window of 1.4 $m/z$; resolution of 30,000; injection time of 128 ms; and automatic gain control target of $2 \times 10^5$.

Result raw files were converted to the mgf format with ProteomeDiscoverer 2.1.0.81 (Thermo Fisher Scientific): signal-to-noise ratio of 1.5, and precursor mass of 350–7,000 Da. Crosslinked peptides were identified with pLink v2.3.9 (pFind group[44]) and the following parameters: missed cleavage sites was 3; fixed modification was carbamidomethylation of cysteines; variable modification was oxidation of methionines; peptide tolerance was 10 p.p.m.; fragment tolerance was 20 p.p.m.; peptide length was 5–60 amino acids; and the spectral false discovery rate was 1%. The sequence database was assembled from all proteins within the complex. Crosslink sites were visualized with XiNet[45] and the Xlink Analyzer[46] plugin in Chimera.

The samples for the ubiquitylation analysis were produced by an in vitro ubiquitylation assay as described above. Control sample was prepared in the same way but without the addition of ubiquitin to make sure that endogenously purified Pol II was not already ubiquitylated. In addition to site-specific Pol II ubiquitylation, promiscuous ubiquitylation of free CSB and UVSSA was observed that probably resulted from a population of TCR factors not bound to Pol II.

For mass spectrometry, the samples were reduced with 5 mM DTT for 30 min at 37 °C and alkylated with 20 mM chloroacetamide for 30 min at room temperature. Unreacted chloroacetamide was quenched by supplementing an additional 5 mM DTT. Proteolytic digestion was performed overnight in denaturing conditions (1 M urea) with trypsin (Promega) in a 1:20 (w/w) protein ratio. The digestion mixtures were acidified with FA to 1% (v/v) end concentration and ACN was added to 5% (v/v) final concentration. Reversed-phase chromatographical purification for mass spectrometric analysis was performed with Harvard Apparatus Micro SpinColumns C18 by washing away salts and contaminants with 5% (v/v) ACN and 0.1% (v/v) FA. Purified peptides were eluted with 50% (v/v) ACN and 0.1% (v/v) FA. The peptide mixture was dried under vacuum and resuspended in 2% (v/v) ACN and 0.05% (v/v) TFA (5 µl for 1 µg of estimated protein amount before digestion).

Liquid chromatography with tandem mass spectrometry analysis was performed by injecting 4 µl of the samples in the Dionex UltiMate 3000 UHPLC system (Thermo Fisher Scientific) coupled with the Orbitrap Fusion Tribrid Mass Spectrometer (Thermo Fisher Scientific). Peptides were separated on an in-house packed C18 column (ReproSil-Pur 120 C18-AQ, 1.9-µm pore size, 75-µm inner diameter and 31 cm in length; Dr. Maisch). Chromatographical separation was achieved with 0.1% (v/v) FA (mobile phase A) and 80% (v/v) ACN and 0.08% (v/v) ACN (mobile phase B). A gradient was formed by the increase of mobile phase B from 5% to 42% in 43 min. Eluting peptides were analysed by data-dependent acquisition with the following MS1 parameters: resolution of 60,000; scan range of 350–1,500 $m/z$; injection time of 50 ms; and automatic gain control target of $4 \times 10^5$. Analytes with charge states 2–7 were selected for higher-energy collisional dissociation with 30% normalized collision energy. Dynamic exclusion was set to 10 s. Fragment MS2 spectra were acquired with the following settings: isolation window of 1.6 $m/z$; detector type was orbitrap; resolution of 15,000; injection time of 120 ms; and automatic gain control target of $5 \times 10^4$.

The resulting acquisition files were analysed with MaxQuant[47] (v1.6.17.0). Fragment peptide spectra were searched against a database containing all proteins of the complex and common protein contaminants. Oxidation of methionines, acetylation of protein N terminus and ubiquitylation residue on lysines were set as variable modifications. Carbamidomethylation of cysteines was set as a fixed modification. Default settings were used with the following exceptions: main search peptide tolerance was set to 6 p.p.m.; trypsin was selected for digestion enzyme; and maximum missed cleavages were increased to 3.

**Cryo-EM sample preparation and image processing**

The same DNA scaffolds were used for all structures[19]: CGC TCT GCT CCT TCT CCC ATC CTC TCG ATG GCT ATG AGA TCA ACT AG (template strand) and CTA GTT GAT CTC ATA TTT CAT TCC TAC TCA GGA GAA GGA GCA GAG CG (non-template strand). In the case of Pol II complex formation with TCR factors only, the shorter RNA was used: rArUrC rGrArG rArGrG rA. If SPT6, PAF and RTF1 were also present, longer RNA was used: rArCrA rUrCrA rUrArA rCrArU rUrUrG rArArC rArArG rArArU rArUrA rUrUrU rArCrA rArArA rUrCrG rArGrA rGrGrA. The elongation complex was formed as in the RNA extension assays. For the Pol II–CSB–CSA–DDB1–UVSSA structure, the pre-formed elongation complex was mixed with twofold excess of TCR factors in complex formation buffer containing 100 mM NaCl, 20 mM HEPES pH 7.5, 1 mM MgCl$_2$, 4% glycerol and 1 mM DTT. The protein solution was incubated at room temperature for 10 min and purified by the Superose 6 Increase 3.2/300 column equilibrated in complex formation buffer. Peak fractions were crosslinked with 0.1% glutaraldehyde on ice for 10 min and quenched with a mixture of lysine (50 mM final) and aspartate (20 mM final). The quenched protein solution was dialysed in Slide-A-Lyzer MINI Dialysis Device of 20K MWCO (Thermo Fisher Scientific) for 6 h against the complex formation buffer without glycerol. For the Pol II–CSB–CSA–DDB1–UVSSA–ADP•BeF$_3$ structure, the complex was supplemented with 0.5 mM ADP•BeF$_3$ before complex purification by size-exclusion chromatography. In the case of complex formation between Pol II, TCR factors, PAF, SPT6 and RTF1, the pre-formed elongation complex was mixed with twofold excess of all proteins in complex formation buffer. In addition, the reaction was supplemented with P-TEFb and ATP (1 mM final), as previously described[30]. Because ATP was present, we used a CSB ATPase-deficient mutant for complex formation. The complex was incubated at 30 °C for 1 h and purified by a Superose 6 Increase 3.2/300 column equilibrated in complex formation buffer. Downstream steps including crosslinking and dialysis were the same as for the previous samples. Dialysed samples were immediately used for the preparation of cryo-EM grids. Of the sample, 4 µl was applied to glow-discharged R2/1 carbon grids (Quantifoil), which were blotted for 5 s and plunge-frozen in liquid ethane with a Vitrobot Mark IV (FEI) operated at 4 °C and 100% humidity.

Micrographs were acquired on a FEI Titan Krios transmission electron microscope with a K3 summit direct electron detector (Gatan) and a GIF quantum energy filter (Gatan) operated with a slit width of 20 eV. Data collection was automated using SerialEM[48] and micrographs were taken at a magnification of ×81,000 (1.05 Å per pixel) with a dose of 1–1.05 e/Å$^2$ per frame over 40 frames. For Pol II–CSA–DDB1–CSB–UVSSA, a total of 10,300 micrographs were acquired; for Pol II–CSA–DDB1–CSB–UVSSA–ADP•BeF$_3$, 10,940 micrographs were acquired; for Pol II–CSA–DDB1–CSB–UVSSA–SPT6–PAF, 8,365 micrographs were acquired; and for Pol II–CRL4$^{CSA}$–CSB–UVSSA–Spt6–PAF, 19,472 micrographs were acquired. Estimation of the contrast-transfer function, motion correction and particle picking was done on-the-fly using Warp[49]. Initial 2D classification and 3D classification steps were done in CryoSPARC[50], followed by further processing in RELION 3.0 (refs [51–53]). Owing to the flexibility of proteins on the Pol II surface, many rounds of signal subtraction and focused classifications were performed, as detailed for every dataset in Extended Data Figs. 3, 5, 7, 9. As a result, the focused classified maps were assembled into a final composite map for each structure. Masks

were created with UCSF Chimera[54]. The final composite maps were created from focused refined maps and denoised in Warp[49].

## Model building and refinement

The focused refined maps and the final composite maps were used for model building. For the Pol II–CSB–CSA–DDB1–UVSSA structure, we first docked existing structures into the density. An initial CSB model was produced with SWISS-MODEL[55,56] using the Rad26 structure (Protein Data Bank (PDB) code: 5VVR[19]) as the template. The model was fitted into the CSB focused refined map in Chimera[54] and rebuilt in Coot[57], followed by real-space refinement in PHENIX[58]. The CSA–DDB1 crystal structure (PDB code: 4A11 (ref. [14])) was fitted into the CSA–DDB1 focused refined map and real-space refinement in PHENIX[58]. During 3D classifications, the β-propeller B of DDB1 was found to adopt many different conformations, apparently rotating around the junction with the rest of the protein, and the final model reflects the most commonly observed conformation. The N-terminal VHS domain of UVSSA was predicted with SWISS-MODEL[55,56] using the GGA3 VHS domain as a template (PDB code: 1JPL[59]). Guided by the crosslinking mass spectrometry data and EM density, the model was fitted into the CSA–UVSSA focused refined map, followed by several rounds of flexible fitting in Namdinator[60] and real-space refinement in PHENIX[58]. The Pol II model (PDB code: 7B0Y[61]) was fitted into the final map and nucleic acids were modified and built in Coot. All protein models were combined in Coot and real-space refined in PHENIX into the final composite map using secondary structure, base-pairing and base-stacking restrains. For the Pol II–CSB–CSA–DDB1–UVSSA–ADP•BeF$_3$ model, ADP•BeF$_3$ was fitted into the density together with the Pol II–CSB–CSA–DDB1–UVSSA model and real-space refined in PHENIX into the final composite map using secondary structure, base-pairing and base-stacking restrains.

For the Pol II–CSB–CSA–DDB1–UVSSA–SPT6–PAF structure, the SPT6 and PAF models (PDB code: 6TED[21]) were fitted into corresponding focused refined maps, adjusted in Coot and real-space refined in PHENIX. Owing to the improved resolution of the SPT6 core, we built an atomic model for it (the SPT6 core was previously modelled on the backbone level). The C-terminal part of LEO1 was displaced in our structure, and therefore these elements were manually built in Coot and deposited as polyalanine because the register could not be determined with certainty. RNA outside Pol II was poorly resolved, presumably due to the absence of DSIF, so we modelled it on the basis of the previous structure (PDB code: 6TED[21]). All models were combined in Coot and real-space refined in PHENIX in the final composite map. In the case of the Pol II–CSB–CRL4$^{CSA}$–UVSSA–SPT6–PAF complex, 3D classification of the stably bound CSA–DDB1–CSB complex revealed two distinct conformations of CUL4A–RBX1. In the first conformation (state 1), CUL4A interacts with UVSSA; in the second conformation (state 2), CUL4A interacts with CSB. Owing to increased flexibility of CUL4A–RBX1, only a smaller subset of particles was used for the final focused refinement of this region. Both focused refinement rounds yielded reconstructions with well-resolved CSA–DDB1, which was then used to resample maps on the map of CSA–DDB1–CSB reconstructed from all particles with stably bound TCR proteins. The crystal structure of the CUL4A–RBX1 (PDB code: 4A0K[14]) complex was fitted into the corresponding focused refined maps, followed by several rounds of flexible fitting in Namdinator[60] and real-space refinement in PHENIX[58]. The β-propeller B of DDB1 was manually adjusted in Chimera and Coot for both CRL4$^{CSA}$ conformations. The model of Pol II–CSB–CSA–DDB1–UVSSA–SPT6–PAF was combined with CUL4A–RBX1 in Coot and the complete models were real-space refined in corresponding composite maps in PHENIX using secondary structure, base-pairing and base-stacking restrains. For Fig. 3, full RBX1 was modelled on the basis of a CUL4A–RBX1 structure (PDB code: 2HYE[62]) due to lower map quality in this region, and the E2 enzyme–donor ubiquitin complex was not present in the complex and was modelled on the basis of a RNF4 RING–UbcH5a–ubiquitin structure (PDB code: 4AP4[63]). In the case of structures containing a CSB ATPase-deficient mutant, the ATPase lobe 2 of CSB is very flexible. Since the complex was incubated with ATP, it is likely that the structure contains a mixture of empty and ATP-bound CSB molecules, resulting in both pre-translocated and post-translocated states of CSB. Final models were validated in Molprobity[64] and the figures were generated with Chimera[54] and ChimeraX[65].

## Reporting summary

Further information on research design is available in the Nature Research Reporting Summary linked to this paper.

## Data availability

The electron density reconstructions and structure coordinates were deposited to the Electron Microscopy Database (EMDB) and to the PDB under the following accession codes: EMDB-13004 and PDB 7OO3 for structure 1, EMDB-13009 and PDB 7OOB for structure 2, EMDB-13010 and PDB 7OOP for structure 3, EMDB-13015 and PDB 7OPC for structure 4, and EMDB-13016 and PBD 7OPD for structure 5. The crosslinking mass spectrometry data and the ubiquitin mapping data have been deposited to the ProteomeXchange Consortium via PRIDE with the dataset identifier PXD025328.

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

**Acknowledgements** We thank current and former members of the Cramer laboratory, in particular, C. Dienemann for assistance with cryo-EM data collection, S. Vos and Y. Chen for assistance with protein purification, Y. Chen for providing P-TEFb, and C. Diederich and S. Dodonova for insightful discussions; M. Ninov for assistance with mass spectrometry; S. Lorenz for discussions about ubiquitylation; and B. Safaric and M. Scherr for advice on protein labelling. G.K. was supported by a Boehringer Ingelheim Fonds PhD fellowship. H.U. is supported by the Deutsche Forschungsgemeinschaft (SFB860). P.C. was supported by the Deutsche Forschungsgemeinschaft (EXC 2067/1 39072994, SFB860 and SPP2191) and the European Research Council Advanced Investigator Grant CHROMATRANS (grant agreement no. 882357).

**Author contributions** G.K. designed and carried out all of the experiments except for the crosslinking mass spectrometry, which was carried out by A.C. F.R.W. assisted with cryo-EM data processing. H.U. supervised the mass spectrometry. P.C. supervised the research. G.K. and P.C. wrote the manuscript, with input from all authors.

**Funding** Open access funding provided by Max Planck Society.

**Competing interests** The authors declare no competing interests.

**Additional information**
**Correspondence and requests for materials** should be addressed to Patrick Cramer.

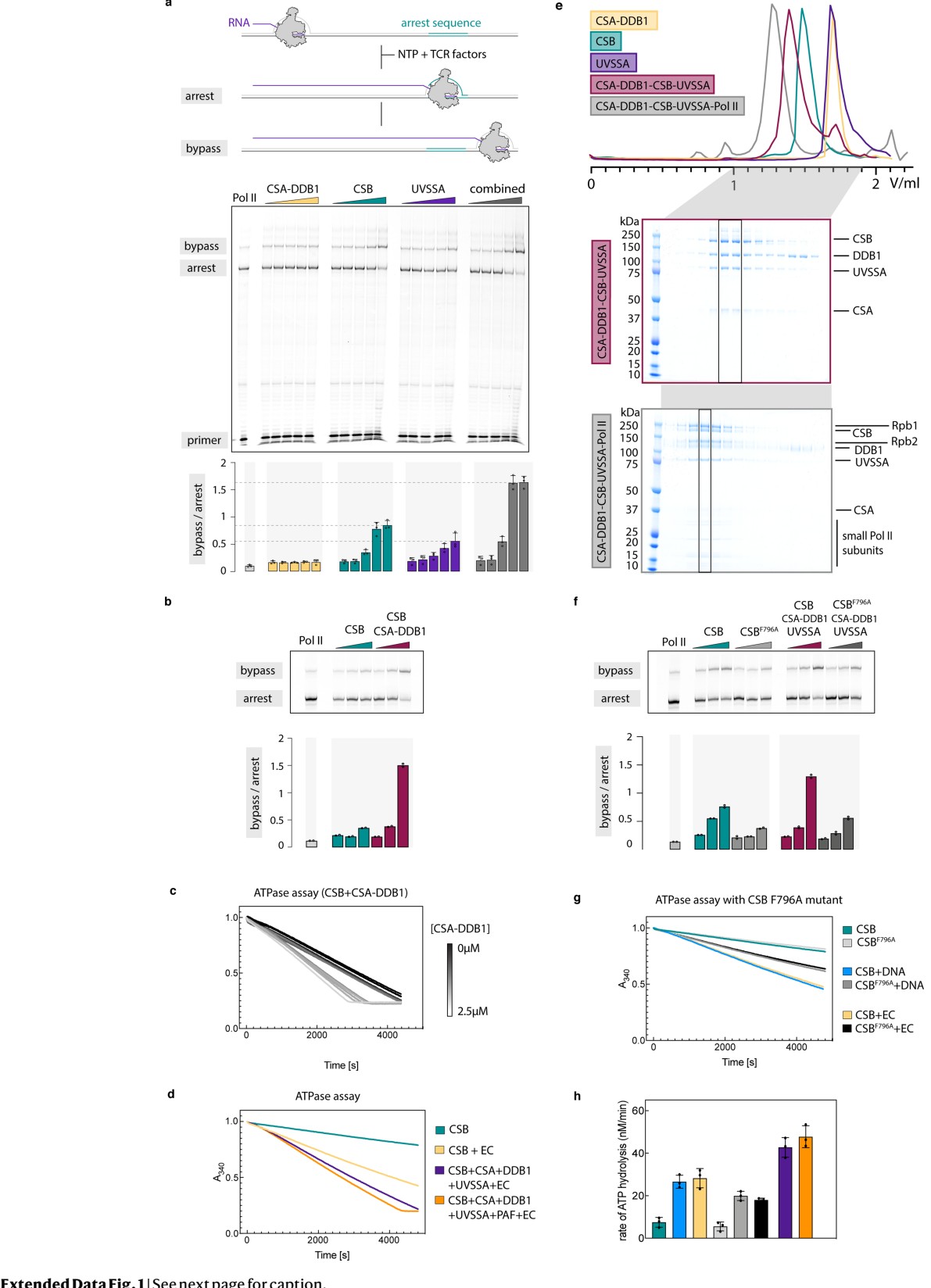

**Extended Data Fig. 1** | See next page for caption.

**Extended Data Fig. 1 | Biochemical characterisation of TCR factors.**
**a**, *In vitro* transcription over an arrest sequence in the presence of TCR factors. The ratio of band intensity for bypass and arrest products was plotted for triplicate measurement. Data are presented as mean values ± SD. **b**, *In vitro* transcription over an arrest sequence in the presence of CSB or CSB-CSA-DDB1 complex. The experiment was repeated two times independently with similar results. Bar graph shows an average value for duplicate measurement. **c**, ATPase assay monitoring CSB activity in the presence of increasing amounts of CSA-DDB1. The experiment was repeated two times independently with similar results. **d**, ATPase assay monitoring stimulation of CSB activity in the presence of Pol II elongation complex, TCR factors and PAF. Analysis shown in **h**. **e**, Analytical size exclusion chromatography of CSA-DDB1, CSB, UVSSA, CSA-DDB1-CSB-UVSSA and Pol II-CSA-DDB1-CSB-UVSSA complexes. The two latter samples were analysed by SDS-PAGE, which confirmed complex composition and purity. The experiment was performed once for individual factors and at least three times for the complexes. **f**, *In vitro* transcription over an arrest sequence in the presence of CSB or CSB mutant F796A. The experiment was repeated two times independently with similar results. Bar graph shows an average value for duplicate measurement. **g**, ATPase assay monitoring the activity of CSB mutant F796A alone, in the presence of bubble DNA and in the presence of a Pol II elongation complex (EC). Analysis shown in **h**. **h**, Summary of ATPase assay results. The rate of ATP hydrolysis was plotted for triplicate measurement. The colour code as in panels **d** and **g**. Data are presented as mean values ± SD. For original gel scans and graph data associated with the Extended Data Fig. 1 see Supplementary Fig. 1 and Supplementary Table 1, respectively.

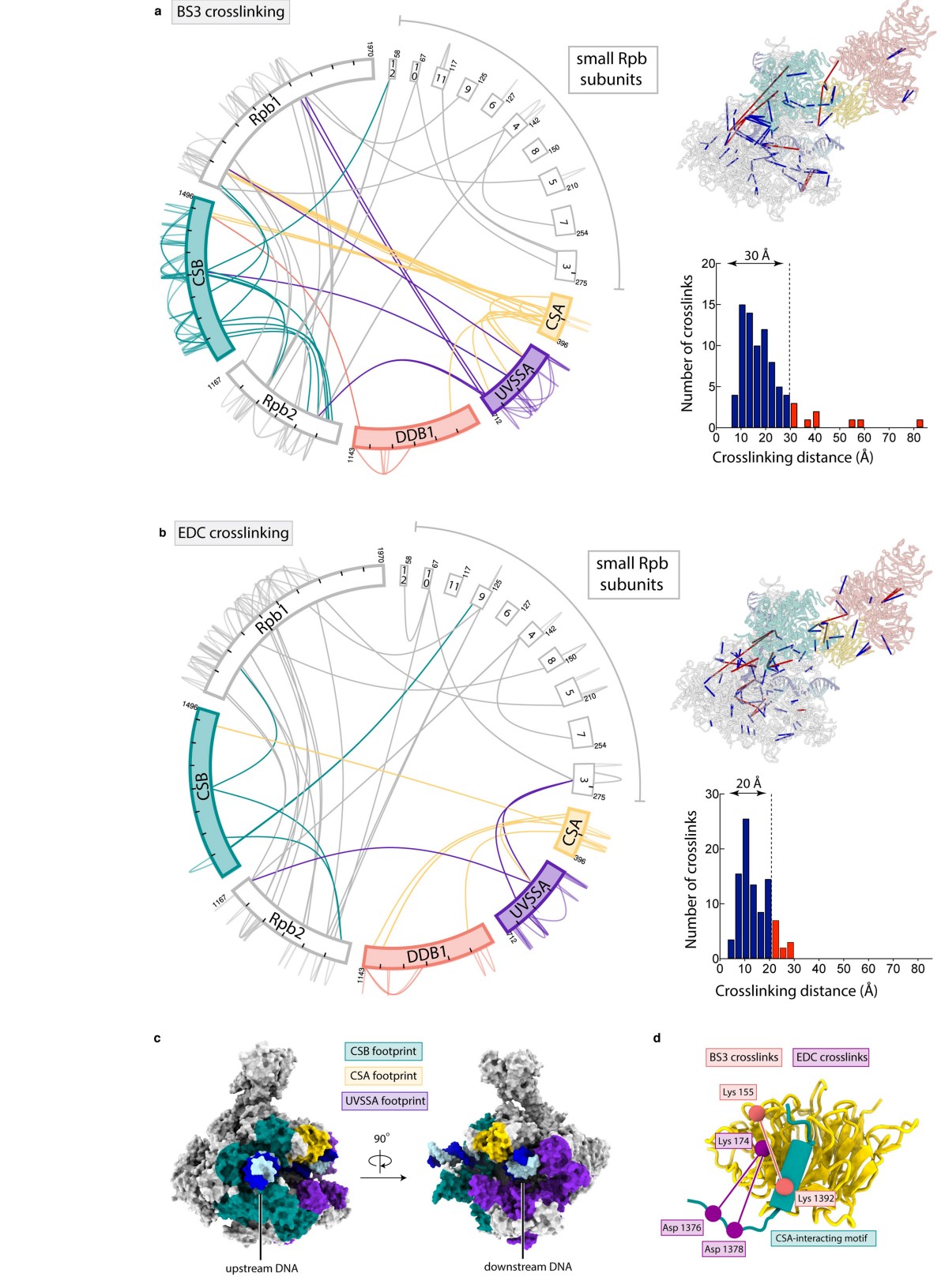

**Extended Data Fig. 2** | See next page for caption.

**Extended Data Fig. 2 | Cross-linking mass-spectrometry interaction networks. a**, Cross-linking mass-spectrometry interaction network within the Pol II-CSA-DDB1-CSB-UVSSA complex after crosslinking with BS3. (right) Crosslinks with the score above 3 that were detected at least twice are shown. (left) Crosslinks were mapped onto the Pol II-CSA-DDB1-CSB structure. Coloured rods connecting crosslinked residues represent permitted (blue) or non-permitted (red) crosslinking distances. 89% of mapped crosslink sites fall within the permitted crosslinking distance of 30 Å. 11% of crosslinks in violation of crosslinking distance are likely a result of complex flexibility or technical errors. Histogram shows the number of crosslinks detected at a particular crosslinking distance. **b**, Cross-linking mass-spectrometry interaction network within the Pol II-CSA-DDB1-CSB-UVSSA complex after crosslinking with EDC. (left) Crosslinks with the score above 3 that were detected at least twice are shown. (right) Crosslinks were mapped onto the Pol II-CSA-DDB1-CSB structure. 83% of mapped crosslink sites fall within the permitted crosslinking distance of 20 Å. 17% of crosslinks in violation of crosslinking distance are likely a result of complex flexibility or technical errors. Histogram shows the number of crosslinks detected at a particular crosslinking distance. **c**, BS3 crosslinks with a score above 3 that were detected at least twice and mapped onto Pol II. The Pol II surface area within the 30 Å radius of the crosslink site was colored as a protein footprint. **d**, BS3 and EDC crosslinks used to identify the CSA-interacting motif (CIM) in CSB.

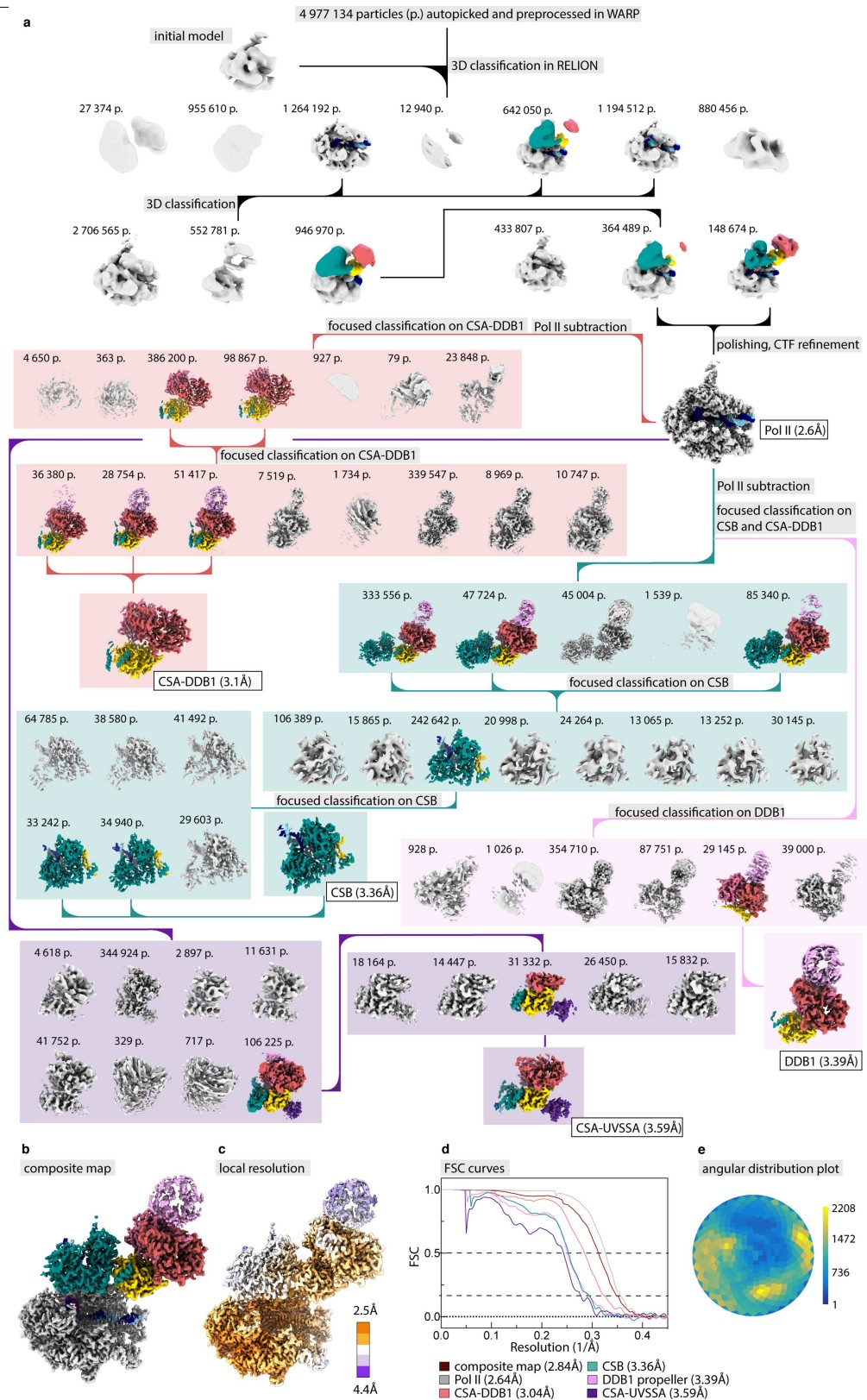

**Extended Data Fig. 3 | Cryo-EM analysis of the Pol II-CSB-CSA-DDB1-UVSSA complex (Structure 1). a**, Processing tree. The number of particles in a particular class is reported above the density. Densities used for further processing are coloured as in Fig. 1a. **b**, Final composite map created from the focused refined maps. **c**, Local resolution estimate for the composite map. **d**, Fourier shell correlation plots for all focused refined maps and the composite map. **e**, Angular distribution plot for the high-resolution Pol II class used as a starting point for focused classifications.

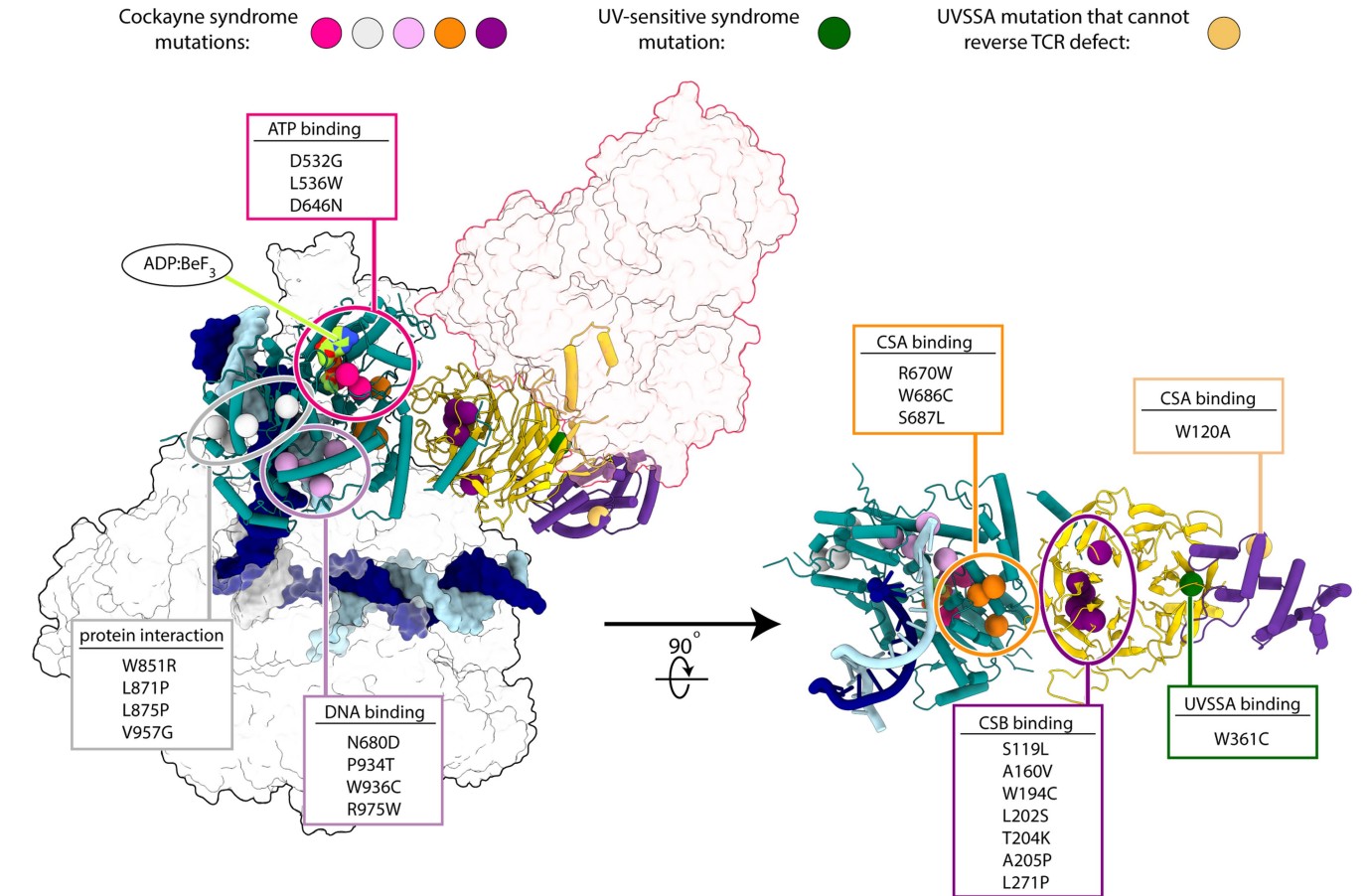

**Extended Data Fig. 4 | Mapping of human disease mutations onto structure 2.** Cockayne syndrome and UV-sensitive syndrome mutations[25] were mapped onto our Pol II-TCR structure and clustered based on their location. A UVSSA mutation that cannot reverse TCR defect was also included[33].

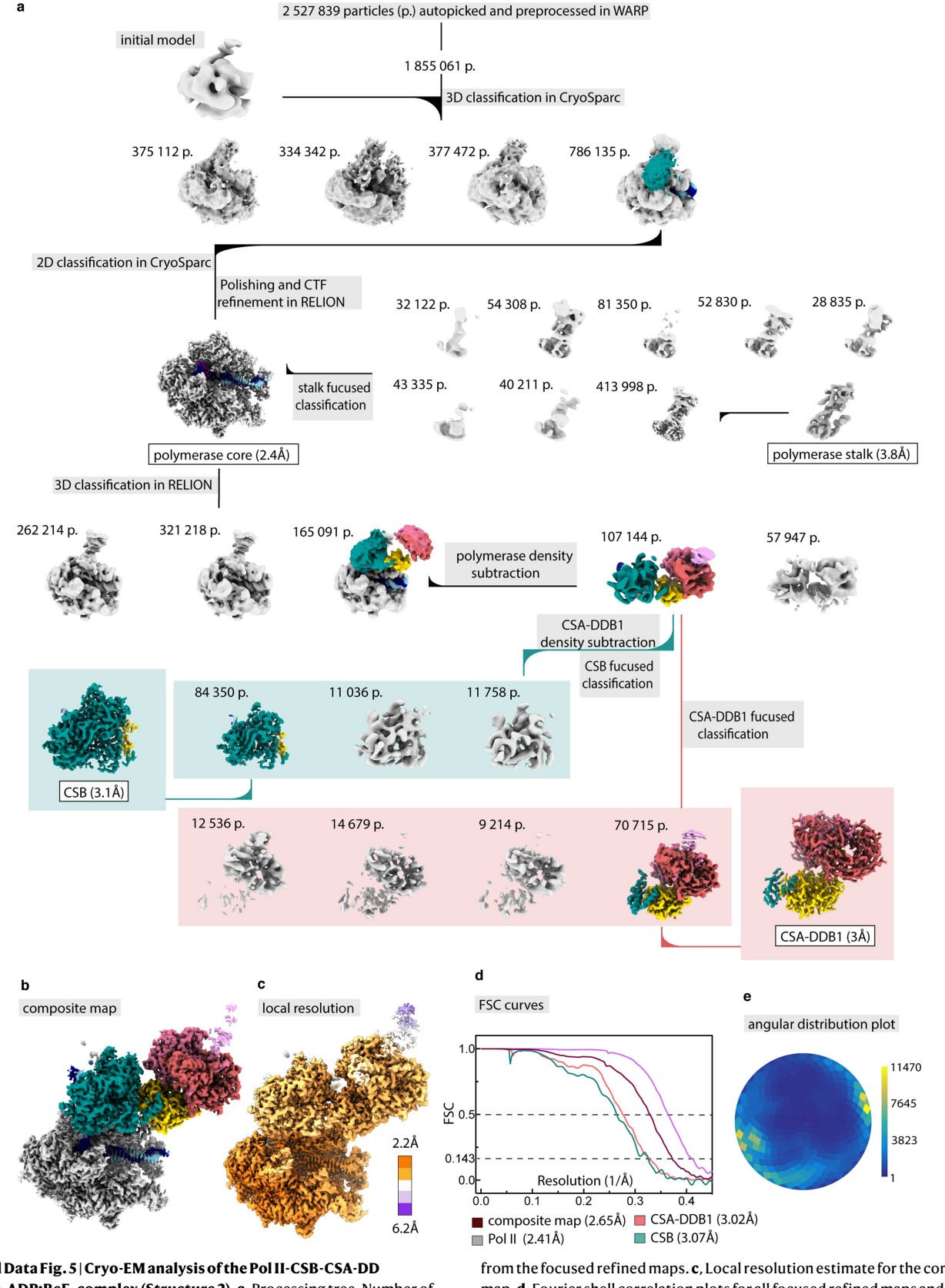

**Extended Data Fig. 5 | Cryo-EM analysis of the Pol II-CSB-CSA-DD B1-UVSSA-ADP:BeF₃ complex (Structure 2). a**, Processing tree. Number of particles in a particular class is reported above the density. Densities used for further processing are coloured as in Fig. 1a. **b**, Final composite map created from the focused refined maps. **c**, Local resolution estimate for the composite map. **d**, Fourier shell correlation plots for all focused refined maps and the composite map. **e**, Angular distribution plot for the high-resolution Pol II class used as a starting point for focused classifications.

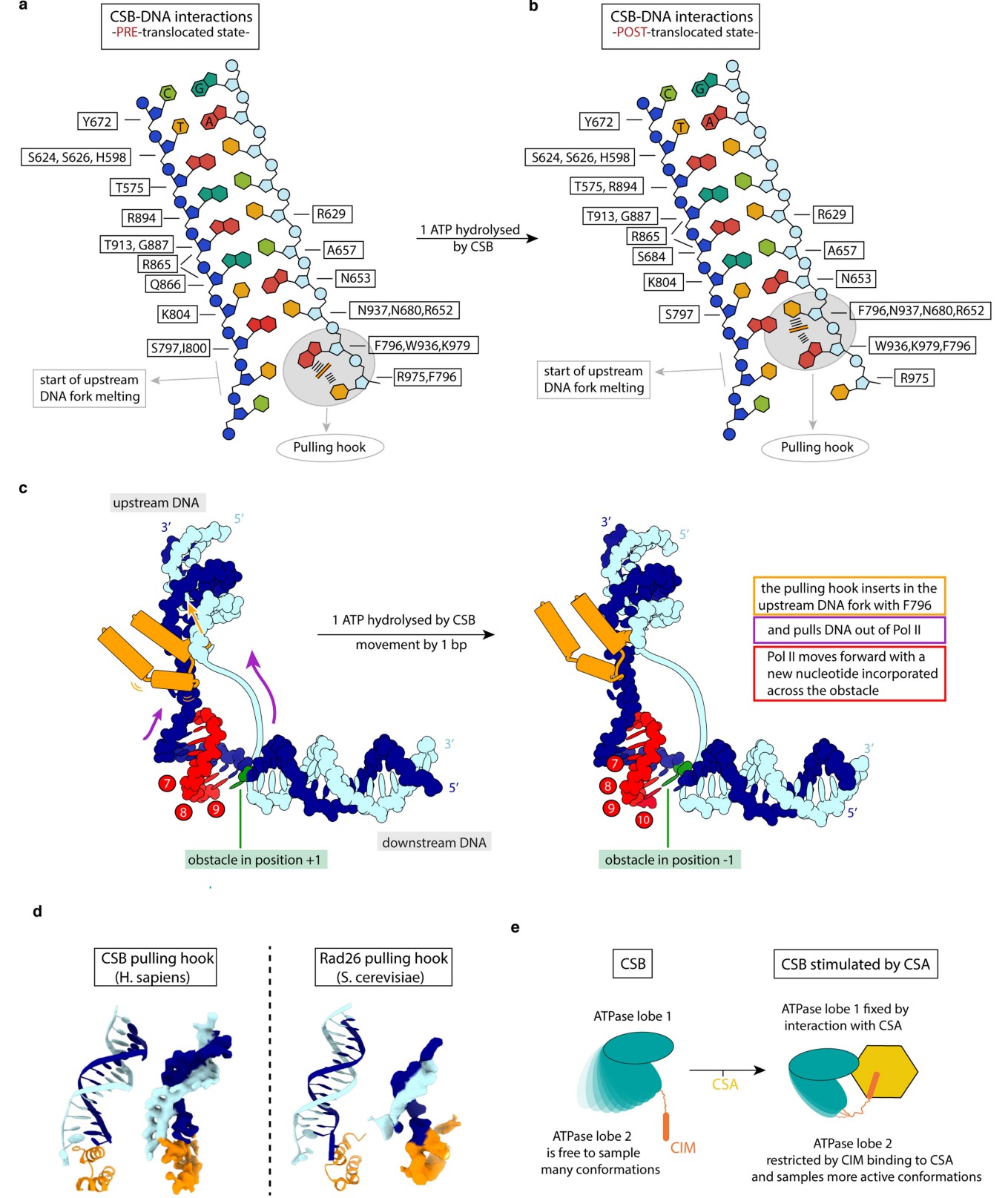

**Extended Data Fig. 6 | DNA binding by CSB and transcription stimulation.**
**a**, CSB interactions with upstream DNA in pre-translocated state (structure 1).
**b**, CSB interactions with upstream DNA in post-translocated state (structure 2).
**c**, Scheme illustrating the movement of DNA during CSB-dependent ATPase

activity. **d**, Comparison of models and cryo-EM densities for the pulling pin in the structure reported here and the yeast Pol II-Rad26 structure reported previously (PDB code 5VVR)[19]. **e**, Structure-based model for CSB stimulation by CSA due to restricting lobe 2 motion via CIM binding to CSA.

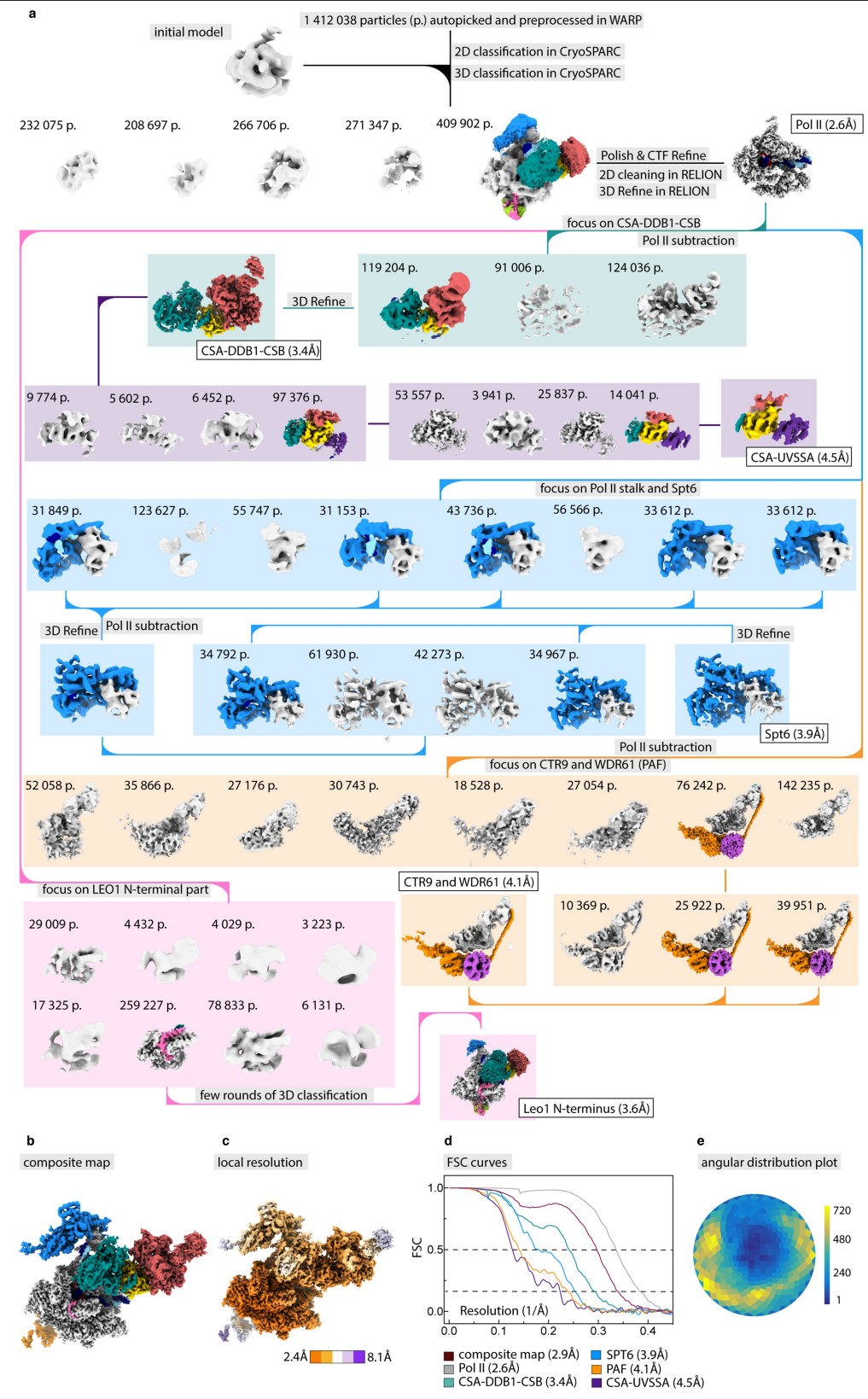

**Extended Data Fig. 7 | Cryo-EM analysis of the Pol II-CSB-CSA-DDB1-UVSSA-SPT6-PAF complex (Structure 3). a**, Processing tree. Number of particles in a particular class is reported above the density. Densities used for further processing are coloured as in Fig. 2c. **b**, Final composite map created from the focused refined maps. **c**, Local resolution estimate for the composite map. **d**, Fourier shell correlation plots for all focused refined maps and the composite map. **e**, Angular distribution plot for the high-resolution Pol II class used as a starting point for focused classifications.

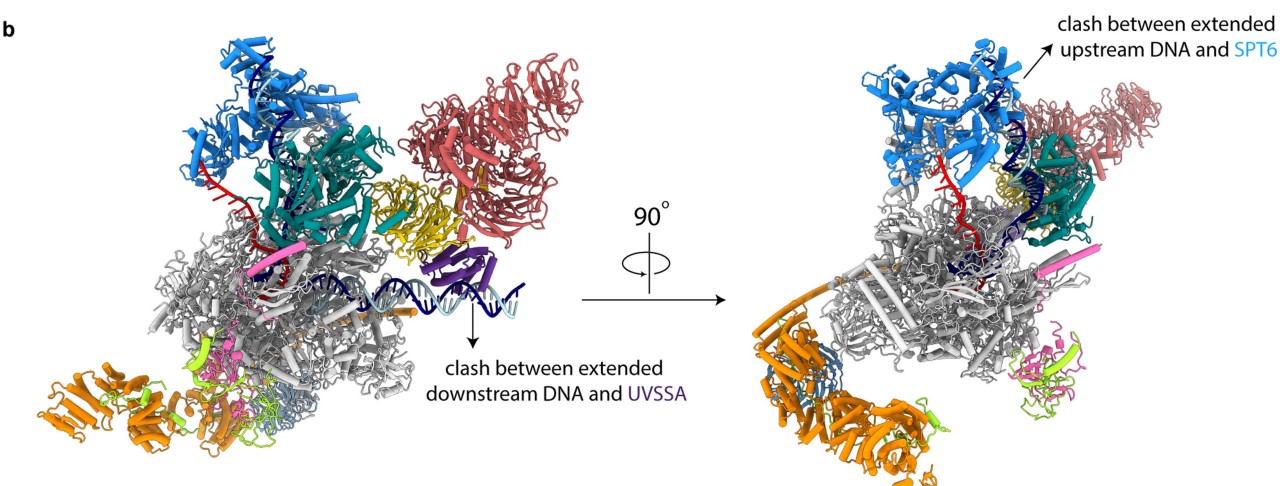

**Extended Data Fig. 8 | Additional analysis of EC^TCR. a**, Analytical size-exclusion chromatography of EC* (left) and EC^TCR (right) in the presence of RTF1. Peak fractions were analysed by SDS PAGE. While RTF1 elutes with EC* in stoichiometric amounts, it elutes with EC^TCR sub-stoichiometrically, which is indicative of weaker association of RTF1 with EC^TCR compared to EC*.

The experiment with EC*-RTF1 was performed once and with EC^TCR-RTF1 twice. For gel source data, see Supplementary Fig. 1. **b**, Modelling shows clashes between UVSSA and extended downstream DNA, and between SPT6 and extended upstream DNA, suggesting that some repositioning of DNA and/or SPT6 and UVSSA occurs when DNA is longer.

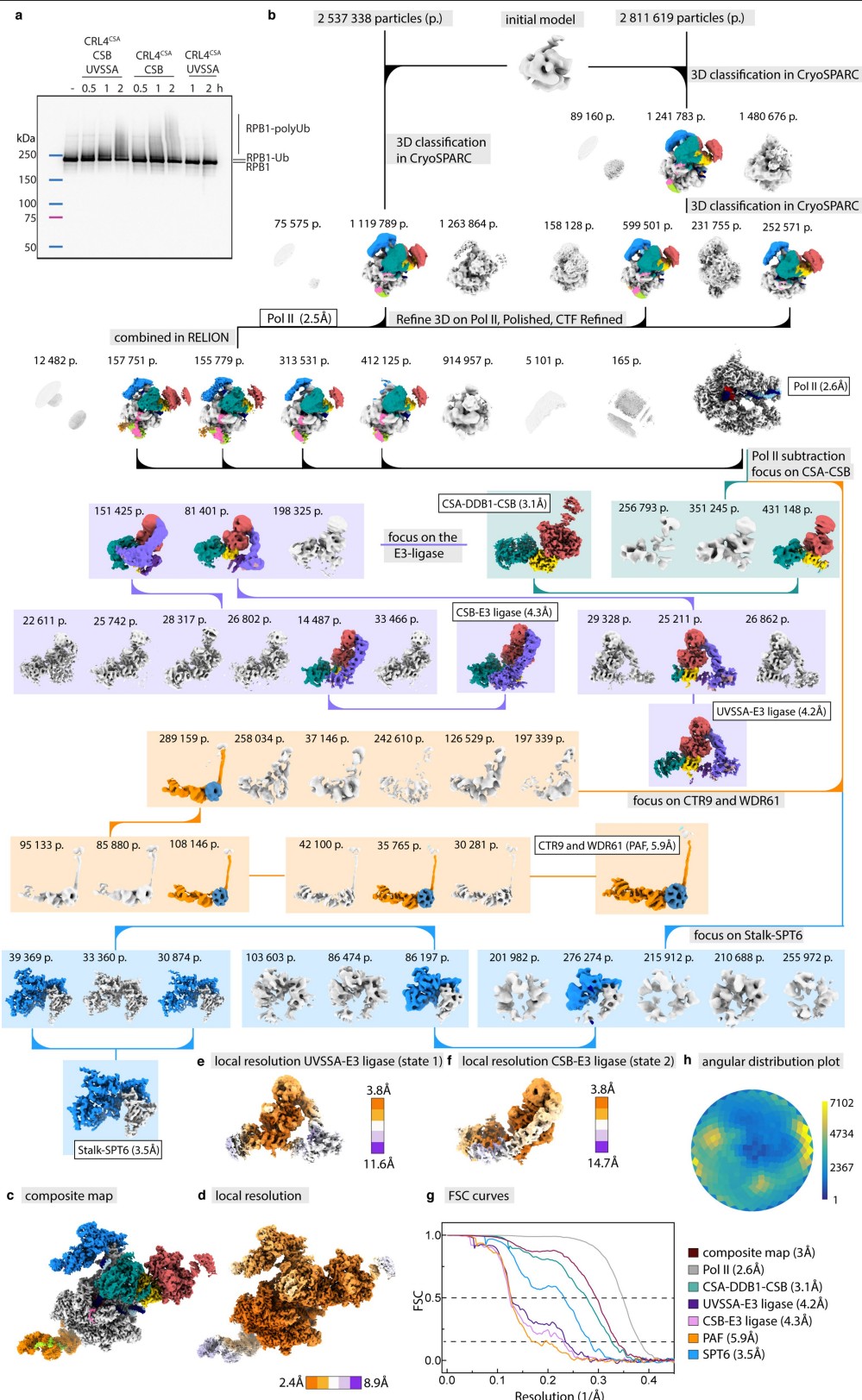

**Extended Data Fig. 9 | Biochemical and cryo-EM analysis of the Pol II-CSB-CRL4$^{CSA}$-UVSSA-SPT6-PAF complex (Structures 4 and 5). a**, Pol II ubiquitination in the absence of UVSSA or CSB. The experiment was repeated two times independently with similar results. For Western blot source data, see Supplementary Fig. 1. **b**, Processing tree. Number of particles in a particular class is reported above the density. Densities used for further processing are coloured as in Fig. 3c. **c**, Final composite map created from the focused refined maps. **d**–**f**, Local resolution estimate for the composite map and focused refined maps containing CRL4$^{CSA}$ **g**, Fourier shell correlation plots for all focused refined maps and the composite map. **h**, Angular distribution plot for the high-resolution Pol II class used as a starting point for focused classifications.

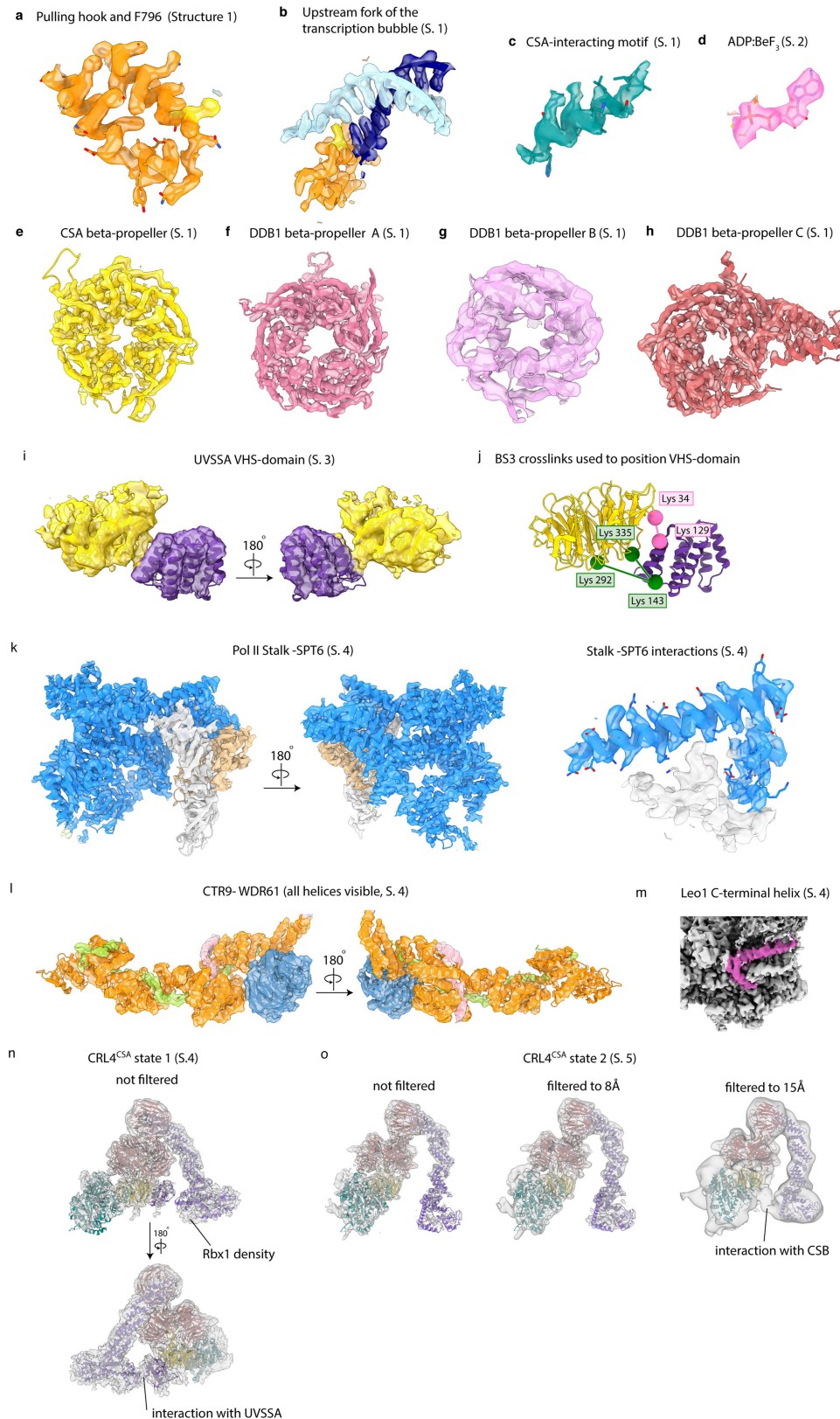

**a**  Pulling hook and F796 (Structure 1)

**b**  Upstream fork of the transcription bubble (S. 1)

**c**  CSA-interacting motif (S. 1)

**d**  ADP:BeF₃ (S. 2)

**e**  CSA beta-propeller (S. 1)

**f**  DDB1 beta-propeller A (S. 1)

**g**  DDB1 beta-propeller B (S. 1)

**h**  DDB1 beta-propeller C (S. 1)

**i**  UVSSA VHS-domain (S. 3)

180°

**j**  BS3 crosslinks used to position VHS-domain

Lys 34
Lys 335
Lys 129
Lys 292
Lys 143

**k**  Pol II Stalk -SPT6 (S. 4)

180°

Stalk -SPT6 interactions (S. 4)

**l**  CTR9- WDR61 (all helices visible, S. 4)

180°

**m**  Leo1 C-terminal helix (S. 4)

**n**  CRL4^CSA state 1 (S.4)
not filtered

180°

Rbx1 density

interaction with UVSSA

**o**  CRL4^CSA state 2 (S. 5)
not filtered    filtered to 8Å

filtered to 15Å

interaction with CSB

**Extended Data Fig. 10 | Quality of cryo-EM densities.** Cryo-EM density of the pulling hook (**a**), pulling hook wedging into the upstream fork of the transcription bubble (**b**), CSA-interacting motif (CIM) in CSB (**c**), ADP:BeF₃ (**d**), beta-propellers in CSA-DDB1 (**e**−**h**), CSA-UVSSA (**i**) together with crosslinks used to position the VHS-domain of UVSSA (**j**), SPT6 and a zoom-in on the SPT6-Pol II stalk contacts (**k**), CTR9 and WDR61 (**l**), LEO1 C-terminal helix (**m**), CRL4^CSA conformation targeting Pol II (**n**), CRL4^CSA conformation targeting CSB (**o**). For each density we indicated from which structure the density was taken, Structure 1, 2, 3, 4 or 5.

**Extended Data Table 1 | Cryo-EM data collection, refinement and validation statistics**

| | Structure 1 (PDB 7OO3) (EMDB-13004) | Structure 2 (PDB 7OOB) (EMDB-13009) | Structure 3 (PDB 7OOP) (EMDB-13010) | Structure 4 (PDB 7OPC) (EMDB-13015) | Structure 5 (PDB 7OPD) (EMDB-13016) |
|---|---|---|---|---|---|
| Magnification | | | 81 000x | | |
| Voltage (kV) | | | 300 | | |
| Electron exposure (e–/$\mathring{A}^2$) | 40.4 | 41.2 | 40.4 | 40.4 | 40.4 |
| Defocus range (μm) | | | 0.5-2 | | |
| Pixel size ($\mathring{A}$) | | | 1.05 | | |
| Symmetry imposed | | | C1 | | |
| Initial particle images (no.) | 4 977 134 | 2 527 839 | 1 412 038 | 5 348 957 | |
| Map resolution ($\mathring{A}$, FSC 0.143) | 2.8 | 2.7 | 2.9 | 3 | |
| Map resolution range ($\mathring{A}$) | 2.5 – 4.4 | 2.2 – 6.2 | 2.4-8.1 | 2.4-8.9 | |
| Map sharpening $B$ factor ($\mathring{A}^2$) | | Denoised map with flattened amplitude spectrum | | | |
| **Refinement** | | | | | |
| Initial model (PDB code) | | | | | |
| Model resolution ($\mathring{A}$) | 2.8 | 2.6 | 2.9 | 2.8 | 2.8 |
| FSC threshold | 0.5 | 0.5 | 0.5 | 0.5 | 0.5 |
| **Model composition** | | | | | |
| Non-hydrogen atoms | 49 391 | 46 263 | 62 442 | 64 227 | 64 231 |
| Protein residues | 6055 | 5550 | 8426 | 9144 | 9144 |
| Nucleotide | 92 | 92 | 103 | 103 | 103 |
| Ligands | 8 Zn, 1 Mg | 8 Zn, 2 Mg, ADP:BeF$_3$ | 8 Zn, 1 Mg | 8 Zn, 1 Mg | 8 Zn, 1 Mg |
| **$B$ factors ($\mathring{A}^2$)** | | | | | |
| Protein | 43.2 | 29.6 | 68.7 | 64.5 | 65.3 |
| Ligand | 72.6 | 32.5 | 71.8 | 90.5 | 91.3 |
| **R.m.s. deviations** | | | | | |
| Bond lengths ($\mathring{A}$) | 0.009 | 0.005 | 0.007 | 0.017 | 0.019 |
| Bond angles (°) | 0.902 | 0.759 | 0.922 | 1.033 | 1.065 |
| **Validation** | | | | | |
| MolProbity score | 1.5 | 1.4 | 1.6 | 1.7 | 1.7 |
| Clashscore | 4.5 | 3.2 | 4.9 | 5.3 | 5.3 |
| Poor rotamers (%) | 0 | 0 | 0 | 0.05 | 0.04 |
| **Ramachandran plot** | | | | | |
| Favored (%) | 96.2 | 96 | 95.2 | 94.12 | 94.05 |
| Allowed (%) | 3.8 | 4 | 4.7 | 5.79 | 5.86 |
| Disallowed (%) | 0 | 0 | 0.1 | 0.09 | 0.09 |

# nature research

# Reporting Summary

Nature Research wishes to improve the reproducibility of the work that we publish. This form provides structure for consistency and transparency in reporting. For further information on Nature Research policies, see our Editorial Policies and the Editorial Policy Checklist.

## Statistics

For all statistical analyses, confirm that the following items are present in the figure legend, table legend, main text, or Methods section.

| n/a | Confirmed | |
|---|---|---|
| ☐ | ☒ | The exact sample size (*n*) for each experimental group/condition, given as a discrete number and unit of measurement |
| ☒ | ☐ | A statement on whether measurements were taken from distinct samples or whether the same sample was measured repeatedly |
| ☒ | ☐ | The statistical test(s) used AND whether they are one- or two-sided<br>*Only common tests should be described solely by name; describe more complex techniques in the Methods section.* |
| ☒ | ☐ | A description of all covariates tested |
| ☒ | ☐ | A description of any assumptions or corrections, such as tests of normality and adjustment for multiple comparisons |
| ☐ | ☒ | A full description of the statistical parameters including central tendency (e.g. means) or other basic estimates (e.g. regression coefficient) AND variation (e.g. standard deviation) or associated estimates of uncertainty (e.g. confidence intervals) |
| ☒ | ☐ | For null hypothesis testing, the test statistic (e.g. *F*, *t*, *r*) with confidence intervals, effect sizes, degrees of freedom and *P* value noted<br>*Give P values as exact values whenever suitable.* |
| ☒ | ☐ | For Bayesian analysis, information on the choice of priors and Markov chain Monte Carlo settings |
| ☒ | ☐ | For hierarchical and complex designs, identification of the appropriate level for tests and full reporting of outcomes |
| ☒ | ☐ | Estimates of effect sizes (e.g. Cohen's *d*, Pearson's *r*), indicating how they were calculated |

*Our web collection on statistics for biologists contains articles on many of the points above.*

## Software and code

Policy information about availability of computer code

| Data collection | Serial EM 3.8 beta 8 |
|---|---|
| Data analysis | RELION 3.0 beta-2, UCSF Chimera 1.13, UCSF ChimeraX v0.8, Coot 0.9, Warp v1.0.7, PHENIX 1.18, cryoSPARC 2.14.2, Prism 9, ImageJ version 1.47v, Molprobity 4.5.1, XlinkAnalyzer version 1.1 |

For manuscripts utilizing custom algorithms or software that are central to the research but not yet described in published literature, software must be made available to editors and reviewers. We strongly encourage code deposition in a community repository (e.g. GitHub). See the Nature Research guidelines for submitting code & software for further information.

## Data

Policy information about availability of data

All manuscripts must include a data availability statement. This statement should provide the following information, where applicable:
- Accession codes, unique identifiers, or web links for publicly available datasets
- A list of figures that have associated raw data
- A description of any restrictions on data availability

The electron density reconstructions and structure coordinates were deposited with the Electron Microscopy Database (EMDB) and with the Protein Data Bank (PDB) under the following accession codes: PDB code 7OO3 and EMDB-13004 for Structure 1, PDB code 7OOB and EMDB-13009 for Structure 2, PDB code 7OOP and EMDB-13010 for Structure 3, PDB code 7OPC and EMDB-13015 for Structure 4 and PBD code 7OPD and EMDB-13016 for Structure 5. The crosslinking mass spectrometric data and the ubiquitin-mapping data have been deposited to the ProteomeXchange Consortium via the PRIDE with the dataset identifier PXD025328 .

# Field-specific reporting

Please select the one below that is the best fit for your research. If you are not sure, read the appropriate sections before making your selection.

☒ Life sciences ☐ Behavioural & social sciences ☐ Ecological, evolutionary & environmental sciences

For a reference copy of the document with all sections, see nature.com/documents/nr-reporting-summary-flat.pdf

# Life sciences study design

All studies must disclose on these points even when the disclosure is negative.

| | |
|---|---|
| Sample size | No statistical methods were used to predetermine sample size. All biochemical experiments were replicated two or more times. Structural data was collected on five independently prepared samples. |
| Data exclusions | No data were excluded from the analyses. |
| Replication | All attempts at replication were successful, at least two repetitions for biochemical assays were performed. Cryo-EM single particle analysis inherently relies on averaging over a large number of independent observations. |
| Randomization | Samples were not allocated to groups. |
| Blinding | Investigators were not blinded during data acquisition and analysis because it is not a common procedure for the methods employed. |

# Reporting for specific materials, systems and methods

We require information from authors about some types of materials, experimental systems and methods used in many studies. Here, indicate whether each material, system or method listed is relevant to your study. If you are not sure if a list item applies to your research, read the appropriate section before selecting a response.

### Materials & experimental systems

| n/a | Involved in the study |
|---|---|
| ☐ | ☒ Antibodies |
| ☐ | ☒ Eukaryotic cell lines |
| ☒ | ☐ Palaeontology and archaeology |
| ☒ | ☐ Animals and other organisms |
| ☒ | ☐ Human research participants |
| ☒ | ☐ Clinical data |
| ☒ | ☐ Dual use research of concern |

### Methods

| n/a | Involved in the study |
|---|---|
| ☒ | ☐ ChIP-seq |
| ☒ | ☐ Flow cytometry |
| ☒ | ☐ MRI-based neuroimaging |

## Antibodies

| | |
|---|---|
| Antibodies used | F-12 Pol II antibody, Santa Cruz Biotechnology, sc-55492; anti-mouse HRP conjugate, Abcam, ab5870 |
| Validation | Mouse monoclonal antibody to RNA polymerase II subunit A. Antibody has been validated for the following applications: EIA, Immunoassay, Precipitation, ELISA, Immunofluorescence, Immunohistochemistry - fixed, Immunoprecipitation, and Western Blot. Here it was used for Western blot.<br><br>Anti-mouse HRP conjugate was used as a secondary antibody. |

## Eukaryotic cell lines

Policy information about cell lines

| | |
|---|---|
| Cell line source(s) | Hi5 cells: Expression Systems, Tni Insect cells in ESF921 media, item 94-002F<br>Sf9 cells: ThermoFisher, Catalogue Number 12659017, Sf9 cells in Sf-9000TM III SFM<br>Sf21 cells: Expression Systems, SF21 insect cells in ESF921 medium, Item 94-003F |
| Authentication | None of the cell lines were authenticated. |
| Mycoplasma contamination | Cell lines were not tested for mycoplasma contamination. |
| Commonly misidentified lines<br>(See ICLAC register) | No commonly misidentified cell lines were used. |

