## [Peer Review File · Nature]

Manuscript Title: Structural basis of human transcription-DNA repair coupling

Reviewer Comments & Author Rebuttals

Reviewer Reports on the Initial Version:

Referees' comments:

Referee #1:

In this study from the Cramer lab, Kokic et al. reconstitute the human transcription-coupled repair complex bound to RNA polymerase II (pol II) and report five high-resolution structures that provide key insights into the positioning of TCR factors onto Pol II, the structural basis of RNA Pol II ubiquitylation, and the conformational changes and functional interplay with Pol II-bound elongation factors. The outstanding features of this work are:

(1) TCR factors are shown to span over the Pol II clamp with CSB contacting upstream DNA, UVSSA close to downstream DNA, and CSA forms a bridge between them. CSA has contacts with CSB and UVSSA on opposite sides of the β -propeller, providing a structural basis for and explaining much of the recruitment data observed in vivo.

(2) The comparison of the structures of Pol II-CSB (structure 1) and Pol II-CSB bound with ADP•BeF3 (structure 2) very nicely show how CSB stimulates the forward translocation of Pol II by the pulling hook containing F796.

(3) Several biochemical approaches and a new structure show that CSB displaces elongation factor DSIF (SPT4/SPT5), thereby switching Pol II from transcription elongation into TCR. In this structure, elongation factor SPT6 contacts DNA due to the CSB-induced bending of upstream DNA, while the LEO1 subunit of PAF1C contacts CSB, consistent with in vivo data.

(4) In vitro ubiquitylation assays with the complete Pol II-TCR complex, including CRL4^{CSA} leads to ubiquitylation of CSA, CUL4A, UVSSA and RPB1 (with K1268 as the highest-scoring site by mass spectrometry). This is further supported by a new structure of Pol II with TCR factors, including CRL4^{CSA} showing that the C-terminus of CUL4A with the active site containing Rbx1 is positioned in close proximity to the K1268 site.

(5) A second conformation of the Pol II-TCR complex, including CRL4-CSA, shows extensive conformational changes enabling CUL4A-Rbx1 to contact the C-terminus of CSB, which is targeted for ubiquitylation. Thus, with CSA as its anchor-point, the CRL4-Rbx1 complex likely targets many component of the TCR complex for ubiquitylation in a manner that must be under tight regulation in vivo.

The manuscripts is well written and builds on the impressive previous work of the authors in the field of transcription regulation. Despite many previous attempts, this study succeeds in reconstituting the human TCR complex bound to Pol II and finally reveals long-awaited structural insights into this process. This is a very exciting development, which is highly relevant not only for the transcription field, but also for the DNA repair and ubiquitylation fields. This is an excellent scientific manuscript with a relevant topic, clarity of hypothesis and logic of the line of reasoning, novelty of the findings, clear references to previous literature, and, and it opens new avenues for future exploration and research. It certainly would deserve being considered for publication in Nature, should the authors appropriately address the following points:

Major points

a) The authors performed in vitro ubiquitylation assays with the complete Pol II-TCR complex and mention that these (line 186) "explain why CSB, CSA and UVSSA are all required for Pol II ubiquitination." Please note that degradation of Pol II is absent in cells deficient in CSA and CSB

(PMID: 22466610, PMID: 8876179), while Pol II degradation is accelerated in UVSSA-deficient cells (PMID: 22466610). This suggests that Pol II ubiquitylation occurs normally in UVSSA-deficient cells, but that accelerated degradation explains why the levels of Pol II ubiquitylation appear lower. To address this issue, it would be helpful if the *in vitro* ubiquitylation assay could be performed with Pol II-CSB-CRL4^{CSA} in the absence of UVSSA (and ideally in the presence of CRL4^{CSA} but in the absence of CSB as a control for specificity) to determine whether UVSSA is required for Pol II ubiquitylation or not. This would show whether UVSSA is required for Pol II ubiquitylation in the reconstituted system employed in this study, which cannot be assessed based on the presented experiments.

b) Abstract line 29; discussion 219-222: The last sentence is rather speculative by suggesting that Pol II can continue to transcribe over repaired DNA following the release of TCR factors. Following the release of early TCR factors (CSB, CSA, UVSSA), the DNA lesions will need to be incised by endonucleases ERCC1-XPF and XPG followed by fill-in by a DNA polymerase. The fill-in will take place in a head-to-head orientation of DNA synthesis respective to the direction of transcription. Given that Pol II is ubiquitylated, it is quite possible that Pol II is removed altogether and degraded, not only as a last resort (PMID: 29699641), but also during successful TCR (PMID: 33685798). Evidence for release of Pol II during has been reported (PMID: 29282293). The structures presented in this study do not provide insight into the fate of Pol II during the later stages of TCR, so I suggest to keep both options open.

c) Lines 96/99: "This supports the view that UVSSA ... Cockayne syndrome compared to UV hypersensitivity syndrome". I propose to tone this down. Mapping CS and UVSS mutations onto the structures is very insightful, but it does not explain the differences between the clinical manifestations of these syndromes. The authors show themselves that the combination of TCR factors, including UVSSA, has an additive effect on *in vitro* transcription across an arrest sequence (Ext. Data Fig. 1a). I am doubtful whether there is convincing evidence that CSB and CSA have a role in transcription elongation. Another explanation is a difference in Pol II ubiquitylation and subsequent degradation, which is abolished in cells deficient in CSB and CSA, but still possible in UVSSA-deficient cells (see (PMID: 22466610, PMID: 32142649, PMID: 33685798). Mutations that affect CRL4CSA recruitment would cause persistent Pol II stalling (causing CS), while mutations that do not affect CRL4CSA recruitment (including the W361C mutation in CSA that affects the UVSSA interaction, but not CSA recruitment itself), would not cause CS but instead cause UVSS.

d) *In vivo* experiments show that UVSSA is the key factor that recruits TFIIH to the Pol II complex, stimulated by the ubiquitylation of Pol II at K1268 (PMID: 32142649; PMID: 32355176). Could the authors speculate more on where TFIIH could be positioned by UVSSA in their current structures also based on their own previous work on TFIIH (PMID: 31253769)?

Minor points

- The Cockayne syndrome proteins are more commonly abbreviated as CSB/CSA rather than CsB/CsA.
- Line 36: Reference 11 only looked at CSB; perhaps it is better to also add refs 7 and 8 (for CSA and UVSSA).
- Line 95/Ext. Data Fig. 4: UV-sensitive syndrome (UVSS) rather than UV hypersensitivity syndrome.
- Fig. 1a: The CSA-binding helix has previously been indicated as the CSA-interacting motif (CIM; PMID: 32355176; PMID: 33685798).
- Ext. Data Fig. 4: It may be more clear to indicate W120A as "a UVSSA mutation that causes a TCR defect", rather than "...that cannot reverse a TCR defect".
- Line 127: Mention that DSIF is composed of SPT4/SPT5 for clarity? Especially considering the yeast literature showing mutants of Spt4/Spt5 modulate the requirement for Rad26 in TCR (PMID: 11101522, PMID: 20042611).
- Line 148: I wonder if it is clear that ECTC is an alternative elongation complex containing TCR factors. The authors could consider using ECTCR or something along those lines.

- Line 655: The “Cryo-EM sample preparation and image processing” section refers to TC-NER factors while these are referred to as TCR factors throughout the manuscript.

Referee #2:

The authors have determined cryoEM structures of five mammalian RNA polymerase II (Pol II) – transcription-coupled DNA repair (TCR) complexes. Three of these molecular structures provide valuable insight into how the Cockayne syndrome proteins CsA and CsB and the UV-stimulated scaffold protein A (UVSSA) interact with Pol II, and the interplay of these proteins with transcriptional elongation factors. The other two structures provide new information into the molecular mechanism for the ubiquitination of Pol II or CsB. The five structures appear to be well-determined at good resolutions. The authors combine the five cryoEM structures into a model for transcription-coupled DNA repair. This model is the highlight of the manuscript and prominently discussed in the first paragraph, but the model is not tested.

The model is based on what I consider to be the two most significant results of the manuscript. The first is the finding of a CsB ATPase lobe 2 pulling hook that engages the upstream fork of the DNA bubble. The authors find that ATP binding leads to DNA moving with respect to the CsB ATP binding lobe 2. The authors claim this movement from what they call the pre-translocated state (structure 1) to the post-translocated state (structure 2, with ADP•BeF3 bound) pushes Pol II forward to bypass a DNA lesion. The second significant result is the structure of what the authors call the alternative elongation complex, ECact, which shows how CsB displaces the transcriptional elongation factor DSIF, together with repositioning of upstream DNA to engage the Spt6 elongation factor and movement of the PAF1 elongation complex subunit LEO1 to bind to CsB.

There is much to commend regarding the cryoEM structures and the molecular details they provide. However, the key findings of the structures need to be validated through biochemical experiments. For the CsB pulling hook, the authors show that the ATPase activity of a CsB(F796A) pulling hook mutant in a Pol II elongation complex is reduced. This result does not support the authors’ statement that the “the pulling hook is required for CsB function”. It suggests the pulling hook is connected to CsB’s ATPase activity, but it does not demonstrate that the pulling hook is actually required for CsB function since CsB function is more than its ATPase activity. The authors should examine this point by testing the CsB(F796A) mutant in their in vitro transcription bypass assay (Ext. Data Fig. 1a), alone and in the combined complex.

The authors’ TCR model would be far more significant if it were challenged through biochemical experiments. The atomic details provided by the cryoEM structures offers the authors an opportunity to devise experiments to challenge their hypothesis, and I urge them to do so to enhance the impact of this manuscript.

Additional comments:

1. (lines 120-121) The authors need to provide a figure which explicitly and unambiguously shows how ATP binding will push Pol II forward. I have studied Fig. 1 and Ext. Data Fig. 6, and I still cannot convince myself that Pol II will necessarily be pushed forward. The reader needs to be able to visualize this critical point. The new figure could be a new extended figure or incorporated into Ext. Data Fig. 6.
2. (lines 120-121) How far is Pol II pushed forward upon ATP binding? I had assumed the data supports movement by one bp, but the authors do not actually state this. Is there a reason why this is not explicitly stated?
3. (lines 78-79) The authors should validate the structurally observed interactions of CsB with upstream DNA with biochemical experiments (crosslinking is one possibility). The mapping of human disease mutations provides independent support for structure 2 but these mutations alone do not validate the structure.
4. (lines 53-54) The authors state “Together with biochemical probing, our results establish the structural basis for coupling transcription to DNA repair in human cells.” In the absence of experiments to test the authors’ TCR model, this words “establish the” should be replaced with “suggests a”.

5. (line 180-81) The authors state "... UVSSA positions the C-terminal domain of Cul4A such that Rbx1 faces a loop in the RBP1 jaw domain...". The authors should make it clear to the reader that the Rbx1 was modeled and not actually present in their cryoEM structure. In a similar vein, Fig. 3c legend should explicitly state that both Rbx1 and the E2 enzyme-donor ubiquitin complex are modeled and not present in the actual structure. The current language makes it difficult for the reader to determine if Rbx1 and the E2 enzyme-donor ubiquitin complex are present in the structure but perhaps homology modeled or actually absent in the structure.

6. (Ext. Data Fig. 1b) Please explicitly label the two SDS-PAGE gels to make clear that the upper gel is for the CsA-DDB1-CsB-UVSSA complex and the lower gel is for the CsA-DDB1-CsB-UVSSA-Pol II complex. The use of the red and grey outline of the gels does not provide sufficient visual guidance to the reader unless the reader knows to look for this. There is ample space to the left of the two gels for vertical oriented text to identify the complexes.

Referee #3:

In the manuscript "Structural basis of human transcription-DNA repair coupling", Kokic et al. determined five structures of Pol II transcription complexes containing transcription-coupled DNA repair (TCR) factors and elongation factors. The structures at high resolution reveal positions of these factors relative to Pol II, binding of CsB with upstream DNA at near-atomic resolution, displacement of elongation factors by TCR factors, and positioning of ubiquitination enzymes near Pol II for potential reactions. The structures were determined with high quality and nicely presented. The in vitro biochemical analyses were well designed and performed. This is a beautiful work that will benefit the understanding of transcription-coupled DNA repair.

Major scientific concerns:

1. Some of the key points from the current manuscript, especially conclusions centered on the activity of CSB/Rad26, have already been covered in the paper from Dr. Wang Dong's lab (Jun Xu et al., Nature, 2017). Despite that the authors used human proteins, the catalytic function of CSB/Rad26 and its biological context are largely conserved, compromising the conceptual novelty of the current work. In addition, displacement of DSIF is also predictable given the competitive binding sites of CSB/rad26 (Jun Xu et al., Nature, 2017) and DSIF (Nature, 2018, Cramer lab) on Pol II. Therefore, the authors should at least highlight previous findings and state clearly what is new in this manuscript.

2. From the movies, I noticed potential clash of the downstream DNA with UVSSA in all the five structures if the downstream DNA naturally extends, as expected in the genome. Similar clash would exist for the upstream DNA and Spt6 in Structures 3, 4, 5. If the authors use longer DNA with extended upstream and downstream DNA, the positions of UVSSA and Spt6 might be shifted. The authors may predict whether the clashes would exist and, if yes, may provide prediction and discussion of the effect on the structures and the conclusion of this study.

3. The hypothesis about TFIIH looks nice and is possibly correct. The statements in the abstract and main text suggest that it has been proved to be the fact, without supporting evidence from this study. I would suggest the authors tune down the statement, including the presentation in Fi.

4. It remains to be further addressed how could the stalled polymerase get backtracked and how the damaged-DNA is expose to be repaired. This is one of the most important questions in the TCR pathway. I don't think it is necessary to address this question in this study. Making it clear would not weaken the significance of this work but avoid potential misleading to readers and researchers in this field.

4. The authors found that the addition of CsA-DDB1 and UVSSA further stimulates ATPase activity of CsB (Ext. Data Fig. 1). How does the structural model support this point? This needs to be thoroughly discussed.

5. Based on the analysis of disease mutations (several found on the CsB-CsA interface) associated with Cockayne Syndrome, the authors claimed that "CsB and CsA play a more general role in transcription elongation". Although the role of CsB in stimulating elongation is validated in Ext. Data Fig. 1a, it's unclear how CsA contributes to elongation. Does the addition of CsA alone to CsB enhance its elongation-promoting function?

6. SPT5 and RTF1 are absent in the Eact. Although it seems convincing that SPT5 is replaced by CsB as they bind the same site on Pol II, as validated by EMSA (Fig. 2a), I don't think the authors showed enough evidence to prove that RTF1 is displaced. Given that RTF1 loosely associates with PAF1 complex, how do the authors exclude the possibility that RTF1 is absent in this structure due to technical issues? The loss of SPT5 might not be enough for RTF1 dissociation because RTF1 can directly bind PAF1C and Pol II. Moreover, what is the biological consequence by specifically displacing RTF1 over other PAF1C subunits? The displacement of RTF1 needs to be verified and discussed.

Author Rebuttals to Initial Comments:

Responses to reviewer concerns

Kokic et al., Structural basis of human transcription-DNA repair coupling

Responses are in italics.

Referee #1's comments:

In this study from the Cramer lab, Kokic et al. reconstitute the human transcription- coupled repair complex bound to RNA polymerase II (pol II) and report five high- resolution structures that provide key insights into the positioning of TCR factors onto Pol II, the structural basis of RNA Pol II ubiquitylation, and the conformational changes and functional interplay with Pol II-bound elongation factors. The outstanding features of this work are:

- (1) TCR factors are shown to span over the Pol II clamp with CSB contacting upstream DNA, UVSSA close to downstream DNA, and CSA forms a bridge between them. CSA has contacts with CSB and UVSSA on opposite sides of the β -propeller, providing a structural basis for and explaining much of the recruitment data observed in vivo.
- (2) The comparison of the structures of Pol II-CSB (structure 1) and Pol II-CSB bound with ADP•BeF3 (structure 2) very nicely show how CSB stimulates the forward translocation of Pol II by the pulling hook containing F796.
- (3) Several biochemical approaches and a new structure show that CSB displaces elongation factor DSIF (SPT4/SPT5), thereby switching Pol II from transcription elongation into TCR. In this structure, elongation factor SPT6 contacts DNA due to the CSB-induced bending of upstream DNA, while the LEO1 subunit of PAF1C contacts CSB, consistent with in vivo data.
- (4) In vitro ubiquitylation assays with the complete Pol II-TCR complex, including CRL4^{CSA} leads to ubiquitylation of CSA, CUL4A, UVSSA and RPB1 (with K1268 as the highest-scoring site by mass spectrometry). This is further supported by a new structure of Pol II with TCR factors, including CRL4^{CSA} showing that the C-terminus of CUL4A with the active site containing Rbx1 is positioned in close proximity to the K1268 site.
- (5) A second conformation of the Pol II-TCR complex, including CRL4-CSA, shows extensive conformational changes enabling CUL4A-Rbx1 to contact the C-terminus of CSB, which is targeted for ubiquitylation. Thus, with CSA as its anchor-point, the CRL4-Rbx1 complex likely targets many component of the TCR complex for ubiquitylation in a manner that must

be under tight regulation in vivo.

The manuscript is well written and builds on the impressive previous work of the authors in the field of transcription regulation. Despite many previous attempts, this study succeeds in reconstituting the human TCR complex bound to Pol II and finally reveals long-awaited structural insights into this process. This is a very exciting development, which is highly relevant not only for the transcription field, but also for the DNA repair and ubiquitylation fields. This is an excellent scientific manuscript with a relevant topic, clarity of hypothesis and logic of the line of reasoning, novelty of the findings, clear references to previous literature, and, and it opens new avenues for future exploration and research. It certainly would deserve being considered for publication in Nature, should the authors appropriately address the following points:

We thank the reviewer for the nice summary and kind words.

Major points

a) The authors performed in vitro ubiquitylation assays with the complete Pol II-TCR complex and mention that these (line 186) “explain why CSB, CSA and UVSSA are all required for Pol II ubiquitination.” Please note that degradation of Pol II is absent in cells deficient in CSA and CSB (PMID: 22466610, PMID: 8876179), while Pol II degradation is accelerated in UVSSA-deficient cells (PMID: 22466610). This suggests that Pol II ubiquitylation occurs normally in UVSSA-deficient cells, but that accelerated degradation explains why the levels of Pol II ubiquitylation appear lower. To address this issue, it would be helpful if the in vitro ubiquitylation assay could be performed with Pol II-CSB-CRL4^{CSA} in the absence of UVSSA (and ideally in the presence of CRL4^{CSA} but in the absence of CSB as a control for specificity) to determine whether UVSSA is required for Pol II ubiquitylation or not. This would show whether UVSSA is required for Pol II ubiquitylation in the reconstituted system employed in this study, which cannot be assessed based on the presented experiments.

We thank the reviewer for this helpful comment. As the reviewer suggested, we performed the Pol II ubiquitination assay in the absence of UVSSA, as well as in the absence of CSB as a control for specificity. We decided to monitor the reaction by Western blotting against RPB1 due to strong background by other ubiquitinated proteins in the reaction. In the presence of all TCR factors and in the absence of UVSSA we observed rapid mono-ubiquitination of RPB1, followed by ubiquitin chain extension over time. The CSB control did not show RPB1 ubiquitination. We included this figure in the Extended Data Fig. 9a and comment on the result in the text. In light of the new results we removed the sentence pointed out by the reviewer and thank them again for the helpful suggestion.

b) Abstract line 29; discussion 219-222: The last sentence is rather speculative by suggesting that Pol II can continue to transcribe over repaired DNA following the release of TCR factors. Following the release of early TCR factors (CSB, CSA, UVSSA), the DNA lesions will need to be incised by endonucleases ERCC1-XPF and XPG followed by fill-in by a DNA polymerase. The

fill-in will take place in a head-to-head orientation of DNA synthesis respective to the direction of transcription. Given that Pol II is ubiquitylated, it is quite possible that Pol II is removed altogether and degraded, not only as a last resort (PMID: 29699641), but also during successful TCR (PMID: 33685798). Evidence for release of Pol II during has been reported (PMID: 29282293). The structures presented in this study do not provide insight into the fate of Pol II during the later stages of TCR, so I suggest to keep both options open.

We agree that it is currently unknown if Pol II can continue transcription following TCR and we thank the reviewer for pointing this out. We edited the mentioned sentences to ensure it is clearly understood that this part of our model is indeed a speculation.

Lines 96/99: “This supports the view that UVSSA ... Cockayne syndrome compared to UV hypersensitivity syndrome”. I propose to tone this down. Mapping CS and UVSS mutations onto the structures is very insightful, but it does not explain the differences between the clinical manifestations of these syndromes. The authors show themselves that the combination of TCR factors, including UVSSA, has an additive effect on in vitro transcription across an arrest sequence (Ext. Data Fig. 1a). I am doubtful whether there is convincing evidence that CSB and CSA have a role in transcription elongation.

Another explanation is a difference in Pol II ubiquitylation and subsequent degradation, which is abolished in cells deficient in CSB and CSA, but still possible in UVSSA-deficient cells (see (PMID: 22466610, PMID: 32142649, PMID: 33685798).

Mutations that affect CRL4CSA recruitment would cause persistent Pol II stalling (causing CS), while mutations that do not affect CRL4CSA recruitment (including the W361C mutation in CSA that affects the UVSSA interaction, but not CSA recruitment itself), would not cause CS but instead cause UVSS.

We agree with the comments. We have toned down this statement and took into consideration the plausible explanation proposed by the reviewer. We do note that CSA strongly increases CSB-mediated transcription stimulation, as shown by new experiments suggested by reviewer 3, so the additional function for CSB and CSA in elongation remains likely.

c) In vivo experiments show that UVSSA is the key factor that recruits TFIIH to the Pol II complex, stimulated by the ubiquitylation of Pol II at K1268 (PMID: 32142649; PMID: 32355176). Could the authors speculate more on where TFIIH could be positioned by UVSSA in their current structures also based on their own previous work on TFIIH (PMID: 31253769)?

The position of TFIIH within the Pol II-TCR complex is an important question and we were thinking about it carefully. In the end we decided not to discuss it in the current manuscript because several open questions prevent us from proposing a model with high confidence at this point. Although such a model is plausible, it is not known how the DNA lesion is exposed to the repair machinery. Different scenarios could be envisioned and they would require different positionings of TFIIH on the DNA, so we decided to present this step as a cartoon in Fig. 4 to emphasize that it is currently not possible to model this step. However, this question can be addressed in future studies.

Minor points

- The Cockayne syndrome proteins are more commonly abbreviated as CSB/CSA rather than CsB/CsA.

We changed the abbreviations throughout the text and figures.

- Line 36: Reference 11 only looked at CSB; perhaps it is better to also add refs 7 and 8 (for CSA and UVSSA).

We added the references.

- Line 95/Ext. Data Fig. 4: UV-sensitive syndrome (UVSS) rather than UV hypersensitivity syndrome.

We changed the name.

- Fig. 1a: The CSA-binding helix has previously been indicated as the CSA-interacting motif (CIM; PMID: 32355176; PMID: 33685798).

We replaced CSA-binding helix with CSA-interacting motif throughout the text and figures.

- Ext. Data Fig. 4: It may be more clear to indicate W120A as “a UVSSA mutation that causes a TCR defect”, rather than “...that cannot reverse a TCR defect”.

We agree, but the mutation in question is not a mutation found in patients, so writing “a mutation that causes a TCR defect” might be misleading. This mutation was discovered by mutating many conserved residues in UVSSA and trying to complement the TCR defect in UV^S-A cells. Thus, we decided to keep the current annotation.

- Line 127: Mention that DSIF is composed of SPT4/SPT5 for clarity? Especially considering the yeast literature showing mutants of Spt4/Spt5 modulate the requirement for Rad26 in TCR (PMID: 11101522, PMID: 20042611).

We included the composition of DSIF in the text.

- Line 148: I wonder if it is clear that EC^{act} is an alternative elongation complex containing TCR factors. The authors could consider using ECTCR or something along those lines.

We agree with the comment and we decided to use the name suggested by the reviewer. We replaced EC^{act} with EC^{TCR} throughout the text and figures.

• Line 655: The “Cryo-EM sample preparation and image processing” section refers to TC-NER factors while these are referred to as TCR factors throughout the manuscript.

Corrected.

Referee #2:

The authors have determined cryoEM structures of five mammalian RNA polymerase II (Pol II) - transcription-coupled DNA repair (TCR) complexes. Three of these molecular structures provide valuable insight into how the Cockayne syndrome proteins CsA and CsB and the UV-stimulated scaffold protein A (UVSSA) interact with Pol II, and the interplay of these proteins with transcriptional elongation factors. The other two structures provide new information into the molecular mechanism for the ubiquitination of Pol II or CsB. The five structures appear to be well-determined at good resolutions. The authors combine the five cryoEM structures into a model for transcription-coupled DNA repair. This model is the highlight of the manuscript and prominently discussed in the first paragraph, but the model is not tested.

The model is based on what I consider to be the two most significant results of the manuscript. The first is the finding of a CsB ATPase lobe 2 pulling hook that engages the upstream fork of the DNA bubble. The authors find that ATP binding leads to DNA moving with respect to the CsB ATP binding lobe 2. The authors claim this movement from what they call the pre-translocated state (structure 1) to the post-translocated state (structure 2, with ADP•BeF₃ bound) pushes Pol II forward to bypass a DNA lesion. The second significant result is the structure of what the authors call the alternative elongation complex, ECact, which shows how CsB displaces the transcriptional elongation factor DSIF, together with repositioning of upstream DNA to engage the Spt6 elongation factor and movement of the PAF1 elongation complex subunit LEO1 to bind to CsB.

There is much to commend regarding the cryoEM structures and the molecular details they provide. However, the key findings of the structures need to be validated through biochemical experiments. For the CsB pulling hook, the authors show that the ATPase activity of a CsB(F796A) pulling hook mutant in a Pol II elongation complex is reduced. This result does not support the authors' statement that the “the pulling hook is required for CsB function”. It suggests the pulling hook is connected to CsB's ATPase activity, but it does not demonstrate that the pulling hook is actually required for CsB function since CsB function is more than its ATPase activity. The authors should examine this point by testing the CsB(F796A) mutant in their in vitro transcription bypass assay (Ext. Data Fig. 1a), alone and in the combined complex.

We thank the reviewer for the suggestion. As suggested by the reviewer we performed the transcription assay with the CsB pulling hook mutant alone and in the presence of CsA- DDB1 and UVSSA. We observed a defect in transcription stimulation by the CsB mutant compared to its wild-type counterpart in both cases, as now presented in the Extended Data Fig. 1f. This is in agreement with our ATPase assay showing that the CsB pulling hook mutant cannot be activated by the Pol II elongation complex to the extent of the wild type protein (Extended Data Fig. 1g). Thus, the combined data we present now confirm that the pulling hook is required for CsB function and address the reviewer's concern. We would like to thank the reviewer for pointing this out.

The authors' TCR model would be far more significant if it were challenged through biochemical experiments. The atomic details provided by the cryoEM structures offer the authors an opportunity to devise experiments to challenge their hypothesis, and I urge them to do so to enhance the impact of this manuscript.

We are not sure what the reviewer refers to. We feel that our model was challenged to a large extent already by biochemical experiments that we included here, and additional ones we include now for the revision. In particular, we complemented our structural insights using chemical crosslinking and mass spectrometry, protein mutagenesis, ATPase assays, transcription assays, EMSAs with fluorescently labeled components, ubiquitylation assays and ubiquitination site mapping with mass-spectrometry, and all these data are presented in a manuscript together with five very complex high-resolution structures.

During the revision we provide now additional transcription, ubiquitylation and ATPase assays as suggested by the reviewers. Moreover, our structural insights are consistent with published in vivo observations, which further supports our model.

In case the reviewer refers to the question how the lesion is exposed to the nucleotide excision repair machinery, i.e. whether it is true that TFIIH can push Pol II backwards to expose the lesion in downstream DNA, this requires the development of other assays and the stabilization of other complexes for structural analysis and is clearly beyond the scope of our current work. We hope to investigate this further in the future. We edited our model to make absolutely sure the reader understands it is not shown whether TFIIH pushes Pol II backwards.

Additional comments:

1. (lines 120-121) The authors need to provide a figure which explicitly and unambiguously shows how ATP binding will push Pol II forward. I have studied Fig. 1 and Ext. Data Fig. 6, and I still cannot convince myself that Pol II will necessarily be pushed forward. The reader needs to be able to visualize this critical point. The new figure could be a new extended figure or incorporated into Ext. Data Fig. 6.

We updated the Extended Data Fig. 6c in order to further clarify the Pol II stimulation by CSB. In case the reviewer missed it, we note that we had also animated the Pol II stimulation by CSB in the Supplementary Video 2 which presents a molecular movie of the process, and we hope this might help with visualization.

2. (lines 120-121) How far is Pol II pushed forward upon ATP binding? I had assumed the data supports movement by one bp, but the authors do not actually state this. Is there a reason why this is not explicitly stated?

Based on the structure, the switch between pre- and post-translocated state of CSB would lead to Pol II forward movement by one bp. We incorporated this statement into the Extended Data Fig. 6c.

3. (lines 78-79) The authors should validate the structurally observed interactions of CSB with upstream DNA with biochemical experiments (crosslinking is one possibility). The mapping of human disease mutations provides independent support for structure 2 but

these mutations alone do not validate the structure.

In case the reviewer missed it, we indeed performed BS3 and EDC chemical crosslinking on Structure 1 and we could map the crosslinks between CSB and Pol II around the upstream DNA. This is illustrated in the Extended Data Fig. 2c. We now included further labels in that figure for clarification.

4. (lines 53-54) The authors state “Together with biochemical probing, our results establish the structural basis for coupling transcription to DNA repair in human cells.” In the absence of experiments to test the authors’ TCR model, this words “establish the” should be replaced with “suggests a”.

In chemistry, a mechanism is considered to be established when the intermediates of the reaction are structurally known and the path of their interconversion is outlined. We wish to argue that we have done so because we present structures of the key intermediates on the TCR pathway that occur during the switching from active transcription elongation to a state prone to initiate DNA repair. We have also complemented our intermediate structures with biochemical assays, including new assays suggested by reviewers. We have nevertheless changed the word ‘establish’ to ‘provide’ because there are remaining open aspects of the coupling mechanism that can be filled in in the future. We have checked the entire text carefully to make sure the implications of our work have been referred to correctly.

5. (line 180-81) The authors state “... UVSSA positions the C-terminal domain of Cul4A such that Rbx1 faces a loop in the RBP1 jaw domain...”. The authors should make it clear to the reader that the Rbx1 was modeled and not actually present in their cryoEM structure. In a similar vein, Fig. 3c legend should explicitly state that both Rbx1 and the E2 enzyme-donor ubiquitin complex are modeled and not present in the actual structure. The current language makes it difficult for the reader to determine if Rbx1 and the E2 enzyme-donor ubiquitin complex are present in the structure but perhaps homology modeled or actually absent in the structure.

We thank the reviewer for their comment. This was now clarified in the figure legend.

6. (Ext. Data Fig. 1b) Please explicitly label the two SDS-PAGE gels to make clear that the upper gel is for the CsA-DDB1-CsB-UVSSA complex and the lower gel is for the CsA-DDB1-CsB-UVSSA-Pol II complex. The use of the red and grey outline of the gels does not provide sufficient visual guidance to the reader unless the reader knows to look for this. There is ample space to the left of the two gels for vertical oriented text to identify the complexes.

We did so and thank the reviewer for this suggestion.

Referee #3:

In the manuscript “Structural basis of human transcription-DNA repair coupling”, Kocic et al. determined five structures of Pol II transcription complexes containing transcription-coupled DNA repair (TCR) factors and elongation factors. The structures at high resolution reveal positions of these factors relative to Pol II, binding of CsB with upstream DNA at near-atomic resolution, displacement of elongation factors by TCR factors, and positioning of

ubiquitination enzymes near Pol II for potential reactions.

The structures were determined with high quality and nicely presented. The in vitro biochemical analyses were well designed and performed. This is a beautiful work that will benefit the understanding of transcription-coupled DNA repair.

We thank the reviewer for the kind words.

Major scientific concerns:

1. Some of the key points from the current manuscript, especially conclusions centered on the activity of CSB/Rad26, have already been covered in the paper from Dr. Wang Dong's lab (Jun Xu et al., Nature, 2017). Despite that the authors used human proteins, the catalytic function of CSB/Rad26 and its biological context are largely conserved, compromising the conceptual novelty of the current work. In addition, displacement of DSIF is also predictable given the competitive binding sites of CSB/rad26 (Jun Xu et al., Nature, 2017) and DSIF (Nature, 2018, Cramer lab) on Pol II. Therefore, the authors should at least highlight previous findings and state clearly what is new in this manuscript.

We agree and this is why we cite the previous paper several times in our manuscript. We have also changed our description of this previous paper in the introduction to include the aspect of Rad26 DNA pulling already in the introduction (it is also mentioned in the results again). Please note however that Xu et al. presented the Pol II-Rad26 structure at medium resolution (5.8 Å overall resolution) and the Rad26 density was quite fragmented due to protein flexibility, which precluded mechanistic insights such as the crucial interactions with the upstream DNA. We however refrained from pointing this out in detail in the main text. As shown in our Extended Data Figure 6d, the pulling hook in the Rad26 structure is seen not to be inserted into the upstream DNA junction and it is unclear if this is due to a different (inactive) state that was captured or limitations during model building when the protein density remains at medium resolution. Thus, the detailed mechanistic insights into CSB function we provide with high-resolution structures of the two distinct translocation states of CSB are still novel. However we much appreciate the Xu et al. work and double-checked and carefully edited to make sure it is correctly referred to throughout our manuscript.

Regarding CSB-DSIF competition, it was unclear if DSIF is fully displaced because there are DSIF-Pol II interactions that would not be affected by CSB binding, in particular between the SPT5 KOW5 domain and Pol II (Bernecky et al., 2017 NSMB). Thus, the displacement of DSIF by CSB from the Pol II surface is experimentally demonstrated for the first time here.

2. From the movies, I noticed potential clash of the downstream DNA with UVSSA in all the five structures if the downstream DNA naturally extends, as expected in the genome. Similar clash would exist for the upstream DNA and Spt6 in Structures 3, 4, 5. If the authors use longer DNA with extended upstream and downstream DNA, the positions of UVSSA and Spt6 might be shifted. The authors may predict whether the clashes would exist and, if yes, may provide prediction and discussion of the effect on the structures and the conclusion of this study.

We thank the reviewer for noting this. We have described this in the new Extended Data Fig. 8b and comment on it in the main text. This does not change any of our conclusions, however.

3. The hypothesis about TFIIH looks nice and is possibly correct. The statements in the abstract and main text suggest that it has been proved to be the fact, without supporting evidence from this study. I would suggest the authors tune down the statement, including the presentation in Fig. 4. It remains to be further addressed how could the stalled polymerase get backtracked and how the damaged-DNA is exposed to be repaired. This is one of the most important questions in the TCR pathway. I don't think it is necessary to address this question in this study. Making it clear would not weaken the significance of this work but avoid potential misleading to readers and researchers in this field.

We agree with the author that this stage of TCR still remain to be addressed. This is why we deliberately presented the intermediate containing TFIIH as a cartoon drawing to emphasize that the structure of this step is still lacking. We now additionally place '?' next to the stages of TCR cycle that still need to be revealed and we state it clearly in the figure legend. We also carefully edited the abstract to make sure this is not understood to be demonstrated here.

4. The authors found that the addition of CsA-DDB1 and UVSSA further stimulates ATPase activity of CsB (Ext. Data Fig. 1). How does the structural model support this point? This needs to be thoroughly discussed.

Please see the answer to comment 5 below.

5. Based on the analysis of disease mutations (several found on the CsB-CsA interface) associated with Cockayne Syndrome, the authors claimed that "CsB and CsA play a more general role in transcription elongation". Although the role of CsB in stimulating elongation is validated in Ext. Data Fig. 1a, it's unclear how CsA contributes to elongation. Does the addition of CsA alone to CsB enhance its elongation-promoting function?

To address these two comments, we performed more transcription assays with CSB and CSA-DDB1 (Extended Data Fig. 1b). We observed that CSA indeed strongly facilitates CSB-mediated transcription stimulation. We additionally confirmed CSB stimulation by CSA with an ATPase assay (Extended Data Fig. 1c) in which we show that the ATPase activity of CSB is increased by the addition of CSA-DDB1.

Upon further inspection of the CSB-CSA structure we proposed the structural basis for this stimulation, which we now included in the Extended Data Fig. 6e. We also discuss these new findings in the main text as suggested by the reviewer. We thank the reviewer for these insightful suggestions that improved our manuscript.

6. SPT5 and RTF1 are absent in the ECact. Although it seems convincing that SPT5 is replaced by CsB as they bind the same site on Pol II, as validated by EMSA (Fig. 2a), I don't think the authors showed enough evidence to prove that RTF1 is displaced. Given that RTF1 loosely associates with PAF1 complex, how do the authors exclude the possibility that RTF1 is absent in this structure due to technical issues? The loss of SPT5 might not be enough for RTF1 dissociation because RTF1 can directly bind PAF1C and Pol II. Moreover, what is the biological consequence by specifically displacing RTF1 over other PAF1C subunits? The displacement of RTF1 needs to be verified and discussed.

To address these concerns we performed the analytical size-exclusion chromatography with EC and EC^{act} in the presence of RTF1. In case of EC* we observed stoichiometric binding of RTF1 to Pol II elongation complex, which is consistent with our previous findings (Vos et al., 2020, Nature). However, in case of EC^{act} we observed that RTF1 still co-elutes with Pol II but in sub-stoichiometric amounts, which indicates a weaker association of RTF1 with EC^{act} compared to EC*. Additional support for our results is the observation that RTF1 stimulation of transcription is significantly weaker in the absence of DSIF (Vos et al., 2020, Nature), as is the case for EC^{act}. Thus, a weaker affinity of RTF1 towards EC^{act} is apparently the reason why we did not observe RTF1 in Structure 3, rather than technical difficulties with cryo-EM. We included this data in the new Extended Data Fig. 8a and we comment on it in the main text.*

Reviewer Reports on the First Revision:

Referees' comments:

Referee #1:

The revised manuscript by Kokic et al. has satisfactorily addressed all my previous points. Ext. Data Fig. 9a elegantly shows that CSB and CRL4^{CSA} are sufficient to catalyze RPB1 ubiquitylation in the reconstituted system in a manner that does not depend on UVSSA. The five presented structures along with the biochemical validation experiments reveal key insights into the coupling between transcription and DNA repair. I highly recommend this manuscript for publication in Nature.

Final small suggestions to consider:

- The authors changed CsA/CsB to CSA/CSB and EC^{act} to EC^{TCR} throughout the text and figures. However, the Supplementary videos still use the old names.
- Line 110: Consider substitution of F796 to alanine rather than truncation considering that this is not a truncating mutation.
- Line 225: "Pol II may now resume transcription and pass over repaired DNA". I see the point, but following the release of CSB, CRL4^{CSA} and UVSSA, the DNA lesion is still not repaired. This will first require the additional binding of core NER and repair synthesis proteins.

Martijn S. Luijsterburg

Referee #2:

The authors have satisfied essentially all of my concerns.

I have one minor suggestion to recommend.

Original item 1. (lines 120-121) The authors need to provide a figure which explicitly and unambiguously shows how ATP binding will push Pol II forward. I have studied Fig. 1 and Ext. Data Fig. 6, and I still cannot convince myself that Pol II will necessarily be pushed forward. The reader needs to be able to visualize this critical point. The new figure could be a new extended figure or incorporated into Ext. Data Fig. 6.

Authors' response: We updated the Extended Data Fig. 6c in order to further clarify the Pol II stimulation by CSB. In case the reviewer missed it, we note that we had also animated the Pol II

stimulation by CSB in the Supplementary Video 2 which presents a molecular movie of the process, and we hope this might help with visualization.

The improved Ext. Data Fig. 6c will better help the reader understand how ATP binding will push Pol II forward. I believe the authors are trying to use colors to match parts of the figures with the written description on the right. However, the different shades (more saturated colors in the figure vs more pastel shades in the written description) makes it unclear to the reader if the colors in the figure are supposed to be paired with the written description. I suggest the authors find a way to use the same colors and shades in the figures and in the written description (Possibility #1: use the pastel shades in the figure. Possibility #2: Draw colored box around each written description instead of using a colored background.)

I should also note that I had viewed Supplementary Video 2 and that was not sufficient to visualize the process. I think the lack of affordances to tell the viewer what to look for or captions makes the video helpful only when the viewer already has a understanding of the mechanism.

Referee #3:

The authors have addressed all my concerns and I have no further major concerns on this revised manuscript. I would like to suggest publication of this work.

Author Rebuttals to First Revision:

Referee #1's comments:

The revised manuscript by Kokic et al. has satisfactorily addressed all my previous points. Ext. Data Fig. 9a elegantly shows that CSA and CRL4CSA are sufficient to catalyze RPB1 ubiquitylation in the reconstituted system in a manner that does not depend on UVSSA. The five presented structures along with the biochemical validation experiments reveal key insights into the coupling between transcription and DNA repair. I highly recommend this manuscript for publication in Nature.

Final small suggestions to consider:

- The authors changed CsA/CsB to CSA/CSB and ECact to ECTCR throughout the text and figures. However, the Supplementary videos still use the old names.

We changed the abbreviations throughout Supplementary videos.

- Line 110: Consider substitution of F796 to alanine rather than truncation considering that this is not a truncating mutation.

We used the wording suggested by the reviewer.

- Line 225: "Pol II may now resume transcription and pass over repaired DNA". I see the point, but following the release of CSB, CRL4CSA and UVSSA, the DNA lesion is still not repaired. This will first require the additional binding of core NER and repair synthesis proteins.

We rephrased the sentence to include that the entire DNA repair process has to occur before transcription may resume.

Martijn S. Luijsterburg

We thank the reviewer for kind words and comments that improved our manuscript.

Referee #2's comments:

The authors have satisfied essentially all of my concerns.

I have one minor suggestion to recommend.

Original item 1. (lines 120-121) The authors need to provide a figure which explicitly and unambiguously shows how ATP binding will push Pol II forward. I have studied Fig. 1 and Ext. Data Fig. 6, and I still cannot convince myself that Pol II will necessarily be pushed forward. The reader needs to be able to visualize this critical point. The new figure could be a new extended figure or incorporated into Ext. Data Fig. 6.

Authors' response: We updated the Extended Data Fig. 6c in order to further clarify the Pol II stimulation by CSB. In case the reviewer missed it, we note that we had also animated the Pol II stimulation by CSB in the Supplementary Video 2 which presents a molecular movie of the process, and we hope this might help with visualization.

The improved Ext. Data Fig. 6c will better help the reader understand how ATP binding will push Pol II forward. I believe the authors are trying to use colors to match parts of the figures with the written description on the right. However, the different shades (more saturated colors in the figure vs more pastel shades in the written description) makes it unclear to the reader if the colors in the figure are supposed to be paired with the written description. I suggest the authors find a way to use the same colors and shades in the figures and in the written description (Possibility #1: use the pastel shades in the figure. Possibility #2: Draw colored box around each written description instead of using a colored background.)

I should also note that I had viewed Supplementary Video 2 and that was not sufficient to visualize the process. I think the lack of affordances to tell the viewer what to look for or captions makes the video helpful only when the viewer already has a understanding of the mechanism.

We thank the reviewer for pointing out this inconsistency. As reviewer suggested we modified the Extended Data Fig. 6c and perfectly matched the color code between the figure and the text.

Referee #3's comments:

The authors have addressed all my concerns and I have no further major concerns on this revised manuscript. I would like to suggest publication of this work.

We thank the reviewer for their time and constructive comments that improved our manuscript.